RESOURCE REPORT
# Construction of an Ordered Transposon Library for Uropathogenic *Proteus mirabilis* HI4320

Melanie M. Pearson,[a] Sapna Pahil,[a]* Valerie S. Forsyth,[a] Allyson E. Shea,[a] Harry L. T. Mobley[a]

aDepartment of Microbiology and Immunology, University of Michigan, Ann Arbor, Michigan, USA

Melanie M. Pearson and Sapna Pahil contributed equally to this article. Author order was determined by seniority.

**ABSTRACT** Ordered transposon libraries are a valuable resource for many bacterial species, especially those with difficult methods for generating targeted genetic mutations. Here, we present the construction of an ordered transposon library for the bacterial urinary tract pathogen *Proteus mirabilis* strain HI4320. This library will facilitate future studies into *P. mirabilis* biology. For large experimental screens, it may be used to overcome bottleneck constraints and avoid biased outcomes resulting from gene length. For smaller studies, the library allows sidestepping the laborious construction of single targeted mutants. This library, containing 18,432 wells, was condensed into a smaller library containing 1,728 mutants. Each selected mutant had a single transposon insertion in an open reading frame, covering 45% of predicted genes encoded by *P. mirabilis* HI4320. This coverage was lower than expected and was due both to library wells with no mapped insertions and a surprisingly high proportion of mixed clones and multiple transposon insertion events. We offer recommendations for improving future library construction and suggestions for how to use this *P. mirabilis* library resource.

**IMPORTANCE** Ordered libraries facilitate large genetic screens by guaranteeing high genomic coverage with a minimal number of mutants, and they can save time and effort by reducing the need to construct targeted mutations. This resource is now available for *P. mirabilis*, a common and complicating agent of catheter-associated urinary tract infection. We also present obstacles encountered during library construction with the goal to aid others who would like to construct ordered transposon libraries in other species.

**KEYWORDS** *Proteus mirabilis*, mutagenesis, ordered library, transposons, urinary tract infection

Disruption of single genes has been a cornerstone of molecular bacteriology. However, this process is time consuming and impractical if many genes are to be studied simultaneously. In some cases, dedicated teams have systematically mutated or deleted every nonessential gene to create mutant libraries (1–4). This is not always feasible depending on the organism being studied, the ease of generating specific mutations or deletions, and the availability or cost of personnel to construct thousands of mutants.

To this end, random transposon mutagenesis allows for the simultaneous screening of tens of thousands of mutants. However, this method has historically been limited by the relative difficulty in determining the precise genomic location of transposon insertions and the capacity of screening experiments to handle large numbers of mutants (e.g., laborious screens or screens with bottleneck effects). The combination of random mutagenesis with modern high-throughput sequencing has facilitated the development of ordered transposon libraries where most nonessential genes in an organism are available as single-insertion mutants in an array of 96-well plates (for example, see references 5–12). The resulting ordered library allows the selection of specific mutants, and any interesting mutants obtained from a screen using an ordered library can be easily gathered for follow-up studies. Assembly of ordered libraries

Address correspondence to Melanie M. Pearson, mpears@umich.edu.

*Present address: Sapna Pahil, Postgraduate Institute of Medical Education and Research, Chandigarh, India.

The authors declare no conflict of interest.

has been particularly helpful for studying pooled groups of mutants in animal models of infection, especially when models have genetic and physiological bottleneck effects that lead to stochastic loss of mutants (12–15). Thus, condensed libraries containing desirable subsets of mutants can be generated for more focused screening. A condensed library might include, for example, a single transposon insertion per open reading frame or a subset of mutants with similar predicted functions.

To decrease costs associated with sequencing each well of the library individually, algorithms have been developed for pooling, sequencing, and deconvoluting the locations of mutants in ordered libraries (16, 17). One such method, Cartesian Pooling-Coordinate Sequencing (CP-CSeq) (17), leverages Cartesian pooling of arrayed mutants to reduce the number of high-throughput sequencing samples required to accurately identify each individual transposon insertion.

We have now constructed an ordered library using the CP-CSeq method to facilitate studies of *Proteus mirabilis*, a Gram-negative bacterium that is a member of the order *Enterobacterales*. *P. mirabilis* is a widely distributed species in the environment, an occasional resident of the human intestinal microbiota, and an opportunistic pathogen (18, 19). Clinically, it is most commonly identified as a cause of complicated urinary tract infections (UTIs), particularly in patients with long-term indwelling urinary catheters (20–22). During these infections, *P. mirabilis* produces urease, which leads to an increase in urinary pH and the precipitation of minerals, resulting in problematic urinary stones and catheter blockage (23).

Traditional targeted mutagenesis in *P. mirabilis* is often time consuming and labor intensive, relying either on individually synthesized group II intron "targetrons" or allelic exchange methods (24, 25). We anticipate this species will be subject to similar bottleneck issues to those found for uropathogenic *Escherichia coli* in our mouse model of ascending UTI, where a maximum of 500 mutants can be screened without random loss of some strains (12). For these reasons, we aimed to generate an ordered library in *P. mirabilis* HI4320, a strain originally isolated from a patient with a long-term indwelling urinary catheter (26, 27) and now considered a reference strain for the species (28). Here, we describe the construction of an ordered library in *P. mirabilis* HI4320 that can be used directly or subdivided into specific topics of interest. We present one such subset as a collection of single insertions in 1,728 genes. We also discuss the limitations and difficulties we encountered in the making of this library.

## RESULTS

**Construction of *P. mirabilis* ordered library.** We used the well-characterized *mariner Himar1* transposon to generate 18,432 mutants of *P. mirabilis* HI4320 arrayed in 192 96-well plates and then used the CP-Cseq method to identify the transposon insertion sites in each well (17). This technique can distinguish transposon insertions across 96 96-well plates. In CP-Cseq, samples are pooled across rows, columns, and plates such that 9,216 wells of an arrayed library are assembled into 40 pools for sequencing. Each well of the library is represented in four pools that are bioinformatically deconvoluted to derive the location of each mutant. Transposon insertions that are identified in one of each of the four types of sequencing pools can be precisely mapped to a well in the ordered library. Insertions that map to more than four pools can be mapped with various degrees of certainty. This might occur, for example, if an identical insertion is located in more than one well. The metrics for all sequencing pools are shown in Table S1 in the supplemental material.

Historically, we have found 58% coverage of predicted genes for the 5.2-MB *E. coli* CFT073 genome using this method and (using 192 96-well plates) 72% coverage of the 5.5-MB *K. pneumoniae* KPPR1 genome (11, 12). The *P. mirabilis* HI4320 genome is substantially smaller (4.1 MB) than either *E. coli* or *K. pneumoniae*, so we expected to have an even higher proportion of genes with transposon insertions in the *P. mirabilis* library. We aimed to achieve a library with an insertion in every, or almost every, gene; therefore, we submitted two sets of plates, designated library 1 (plates 1 to 96) and library 2 (plates 97 to 192), for Illumina sequencing and deconvolution. By modeling the average length of genes in *P. mirabilis* and the average number of TA insertion sites

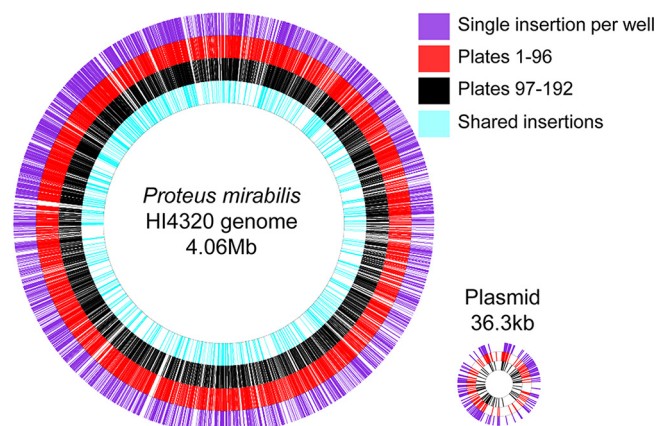

**FIG 1** Mapping of transposon insertions to their respective locations on the chromosome or plasmid. Transposon DNA harvested from a total of 18,432 wells of an ordered library of mutants was sequenced and aligned with the genome or plasmid of HI4320. Each insertion site is marked with a single line. From exterior to interior: location of insertions from wells of the library where a single transposon insertion event was mapped (purple), location of all insertions from plates 1 to 96 of the ordered library (red), location of all insertions from plates 97 to 192 of the ordered library (black), and location of any cases where an insertion at the identical site was identified in both plates 1 to 96 and plates 97 to 192 (light blue). Note that the plasmid is not drawn to scale compared with the chromosome.

per gene, we calculated that 34,561 mutants would be needed to achieve 100% saturation. We then calculated that submitting two libraries would increase our coverage of the *P. mirabilis* HI4320 genome from 90.6% to 99.1% of nonessential genes (Fig. S1).

**Metrics of the ordered library.** *P. mirabilis* HI4320 possesses both a single circular 4.1-MB chromosome and a 36.3-kb plasmid, pHI4320 (28). The combined libraries contained 24,297 transposon insertion events that were mapped to the *P. mirabilis* HI4320 chromosome or plasmid (Table S2). We found the transposon was randomly distributed across both the chromosome and plasmid with similar saturation across both (Fig. 1), consistent with previous Tn-seq experiments using *Himar1* in *P. mirabilis* (29, 30). The full list of transposons mapped to the ordered library can be found in Table S3.

A breakdown of the library contents is shown in Fig. 2. Of the total genome-aligned transposon insertions, 15,091 (62.1%) were identified in exactly four sequencing pools, and therefore, their location could be confidently mapped to the ordered library (Fig. 2A). An additional 1,351 insertions (5.6%) were identified in more than 4 pools and could be partially mapped (e.g., a mutant could be confidently traced to a specific 96-well plate but not a single well) (Table S2).

There were 7,846 wells in the library (42.6%) that were predicted to contain a single transposon insertion; this corresponds to 38.4% with reads in exactly 4 pools and 4.1% with reads in greater than four pools (Fig. 2B). Correspondingly, the number of multiple transposon insertions mapped to a single well was higher than we have observed for comparable previously published libraries in *K. pneumoniae* or *E. coli* from our group (11, 12), particularly for wells with three or more mapped transposons (Table 1). Mapping of multiple transposons to a single well could occur either from multiple transposons inserting in a single genome or because more than one colocalized colony was inadvertently picked into a single well of the library.

After the removal of wells with multiple transposons or low identification certainty, our library contains disruptions in 1,728 open reading frames (45%), of ~3,812 total between the chromosome and plasmid (Fig. 2C). This was lower coverage compared with previous libraries constructed by our group (69% for *K. pneumoniae* KPPR1 and 53% for *E. coli* CFT073), despite the smaller genome for *P. mirabilis* HI4320 (4.06 versus 5.07 and 5.23 Mb for *K. pneumoniae* and *E. coli*, respectively) (11, 12). Notably, the *E. coli* library contained half as many clones as the other two libraries; that is, it contained a single set of 96 96-well plates.

**Validation of Cartesian pooling deconvolution.** Although the CP-Cseq pooling, barcoding, and deconvolution method has been successfully applied to identify transposon insertions in other ordered libraries, we wanted to ensure this was the case for the

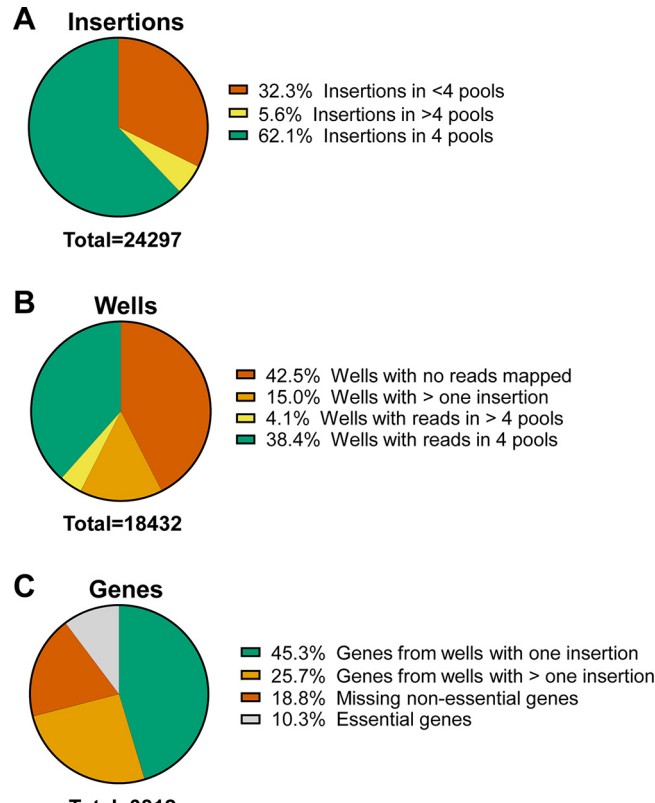

**A Insertions**

- 32.3% Insertions in <4 pools
- 5.6% Insertions in >4 pools
- 62.1% Insertions in 4 pools

**Total=24297**

**B Wells**

- 42.5% Wells with no reads mapped
- 15.0% Wells with > one insertion
- 4.1% Wells with reads in > 4 pools
- 38.4% Wells with reads in 4 pools

**Total=18432**

**C Genes**

- 45.3% Genes from wells with one insertion
- 25.7% Genes from wells with > one insertion
- 18.8% Missing non-essential genes
- 10.3% Essential genes

**Total=3812**

**FIG 2** Composition of the ordered library. Sequencing reads that aligned with *P. mirabilis* HI4320 were then mapped onto the ordered library (A). Insertions that were recovered in exactly four pools could be precisely located in the library. Of the 18,432 wells in the ordered library, 38.4% had a single transposon insertion precisely mapped (B). Of the approximately 3,812 predicted open reading frames encoded by *P. mirabilis* HI4320, 45.3% were represented in the ordered library as a single insertion (C). Genes from wells with more than one insertion often represent mixed clones that could be separated out by single-colony passage. Genes labeled as "essential" would also include transposon insertions that could not be mapped to the *P. mirabilis* chromosome due to repetitive sequence.

*P. mirabilis* library. Therefore, we randomly selected 25 mutants that had single transposon insertions mapped to each well (Table S4).

These 25 mutants were resurrected from the ordered library, and genomic DNA was purified. PCR, using one gene-specific primer and one primer that reads outbound from

**TABLE 1** Wells in the ordered library with multiple transposon insertion events[a]

| Insertions per well | *P. mirabilis*[b] | | *K. pneumoniae*[c] | | *E. coli*[d] | |
|---|---|---|---|---|---|---|
| | No. of wells | Percentage of wells | No. of wells | Percentage of wells | No. of wells | Percentage of wells |
| 1 | 7,846 | 42.6 | 5,889 | 63.9 | 7312 | 79.3 |
| 2 | 1,951 | 10.6 | 496 | 5.4 | 367 | 4.0 |
| 3 | 685 | 3.7 | 75 | 0.81 | 15 | 0.16 |
| 4 | 332 | 1.8 | 12 | 0.13 | 2 | 0.02 |
| 5 | 123 | 0.67 | 3 | 0.03 | 1 | 0.01 |
| 6 | 53 | 0.29 | 2 | 0.02 | | |
| 7 | 26 | 0.14 | | | | |
| 8 | 9 | 0.05 | | | | |
| 9 | 8 | 0.04 | | | | |
| 10 | 3 | 0.02 | | | | |
| 11 | 2 | 0.01 | | | | |

[a]Metrics for the *P. mirabilis* library are compared with previously published libraries in *K. pneumoniae* (11) and *E. coli* (12).
[b]Plates 1 to 192.
[c]Plates 1 to 96 (11).
[d]Plates 1 to 96 (12).

both ends of the transposon, confirmed the transposon insertion in 22/25 of these mutants (87.5%) (Fig. S2). Importantly, this technique only ensured that a mutant was present in a given well of the library, but it would not rule out the presence of unexpected contaminant strains (mixed clones). For the three mutants that did not confirm, two additional gene-specific primers were used to ensure the lack of amplification was not primer specific. Although the vast majority of mutants we tested were validated, this proportion was lower than we have found for other ordered libraries (24/24 for *E. coli*; 13/14 for *K. pneumoniae*) (11, 12) and considerably lower than what was obtained for *Mycobacterium bovis* in the foundational CP-CSeq report (100% of 104 mutants screened) (17). To address the lower-than-expected genome coverage and the many wells predicted to contain multiple transposon insertion events, we designed a series of experiments to explore potential complications with the *P. mirabilis* ordered library.

**Arbitrary PCR revealed unpredicted transposon insertions.** We next reinvestigated the three randomly selected mutants that failed to validate in the initial library PCR screen (Table S4). To determine the actual transposon insertion location(s) in these three outlier mutants, we used arbitrary PCR to amplify transposon-containing sequences (31, 32). This succeeded for one mutant, "PMI2884," which was subsequently confirmed by PCR to have two unpredicted transposon insertions in PMI0378 (*gshA*, glutamine-cysteine ligase) and the intergenic region between PMI1943 and PMI1944. We next used PCR to confirm the predicted insertion in PMI2884 was in the four CP-Cseq sequencing pools that corresponded to this well of the ordered library and found this insertion was indeed present all four pools. This result indicated that in some wells in the library that were deemed to be unique, single-insertion mutants were incorrectly called, because despite repeated attempts we were never able to find a PMI2884 mutant in its expected location in the library. Instead, we confirmed the presence of two insertions in a single clone, neither of which was in the predicted locus determined by Cartesian pooling deconvolution.

**Passaging experiments revealed mixed clones in a single well.** We next considered two explanations for library issues. First, there could be multiple mutants in a single well of the library, which could have occurred during colony picking or due to cross-contamination. Second, the *mariner* transposon might be unstable in *P. mirabilis*. To investigate these possibilities, we looked for restoration of an easily visible phenotype from defined mutants. *P. mirabilis* readily swarms under permissive conditions (33), and we chose this readout to further investigate mixed clones and transposon stability. Two nonmotile mutants containing insertions in flagellar genes *flhD* or *flaA* were picked from the transposon library. Although the *flhD* insertion was predicted to be unique, the *flaA* mutant was selected from a well of the library that was predicted to contain two additional transposons. The *flhD* and *flaA* mutants were single-colony passaged in triplicate into lysogeny broth (LB) and cultured to stationary phase (overnight) and then either streaked for isolation on LB agar or spotted onto swarm agar to assess reversion to wild-type swarming motility. This was serially repeated 10 times. After one passage, one of three *flaA* cultures swarmed (Fig. S3). However, continued passaging of the remaining nonmotile *flhD* or *flaA* mutants yielded no further swarming reversion. We then cultured the primary streak from the original *flaA* resurrected library mutant (in contrast with previous individual colony experiments) and found that it too was motile. We next investigated whether this well of the library, predicted to contain three transposon insertions in a single chromosome, instead contained mixed clones (e.g., three individual strains, each containing a single transposon insertion).

At each passage, samples were collected for genomic DNA extraction. We purified genomic DNA from a variety of motile and nonmotile *flaA* isolates and conducted a Southern blot to assess transposon locations (Fig. 3). It was clear that two transposons moved together with motile isolates (Fig. 3B, lanes 4, 6, and 7), and the third transposon, which moved with nonmotile isolates, was the *flaA* insertion (Fig. 3B, lanes 3, 5, and 8). Indeed, this band was the one that corresponded to the predicted size of the restriction fragment containing *flaA* (2723 bp). This experiment was instrumental in demonstrating that the library has both isolates containing multiple transposons and mixtures of different mutants.

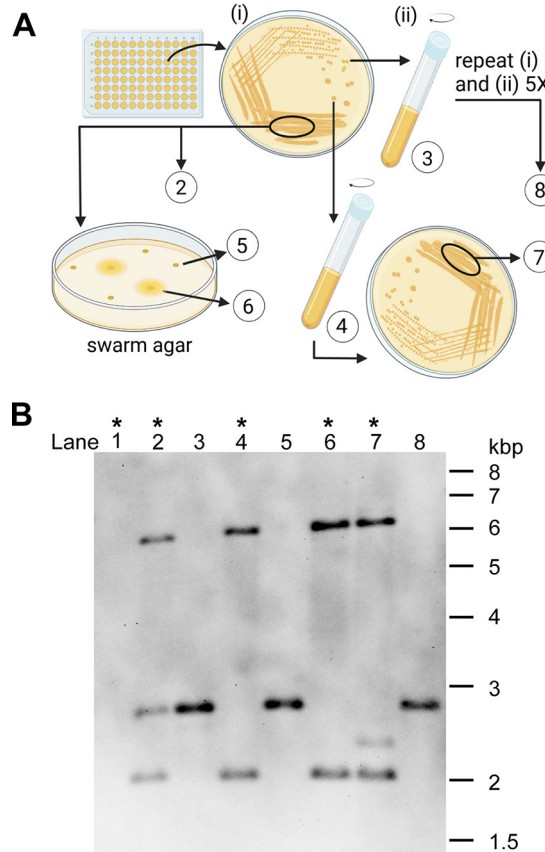

**FIG 3** Three transposons in one well are from two mixed strains. A library well containing three mapped transposons, including *flaA*, was subjected to passaging in triplicate and swarming motility assays. Genomic DNA (gDNA) was purified at the numbered stages (A) and Southern blot was used to track the three insertions (B). Southern blot shows digested gDNA with a probe targeting the kanamycin resistance gene in the transposon. Lane 1 contains wild-type HI4320 (no transposon). All three transposon insertions called by CP-CSeq were present in the well of the library (lane 2). Following passaging, these transposons separated out into two distinct mutants (one containing one transposon and one containing two transposons). The predicted *flaA* band (2,723 bp) tracked with nonmotile isolates and remained stable after five passages (lane 8). Asterisks along the top indicate motile isolates. Size markers are shown on right in kilobase pairs.

**The library contains a substantial proportion of wells with uncalled mixed clones.** We next investigated the accuracy of library wells that had been called to have a single transposon insertion. We selected 7 of the original 25 randomly selected mutants to check for additional transposons using Southern blot. For this experiment, mutants were resurrected onto LB agar from the ordered library, and genomic DNA was isolated from the primary streak instead of isolated colonies. This was done to assay the representative contents of library wells, as opposed to our original PCR confirmation of the predicted single transposon insertion. Southern blot of seven selected mutants showed that five contained more than one band, thus indicating mixed clones or uncalled multiple single-genome insertions were present (Fig. 4A).

To confirm that multiple uncalled transposons were likely to be a result of mixed clones, we single-colony passaged one of the library wells with three bands, the PMI1165 mutant. Genomic DNA was isolated from five colonies, as well as the primary streak of both copies of the full and condensed libraries (described below). Single colony passaging of the "PMI1165" well allowed purification of the single PMI1165 insertion away from the other two insertions (Fig. 4B, colonies 3 to 5). Notably, both copies of the condensed ordered library had an increased proportion of one of the contaminant clones compared with the full library, as assessed by differing relative intensities of each band (Fig. 4B). This result shows that each time the library is manipulated or duplicated, there is a chance for a change in the well population that results in a different ratio of mixed clones.

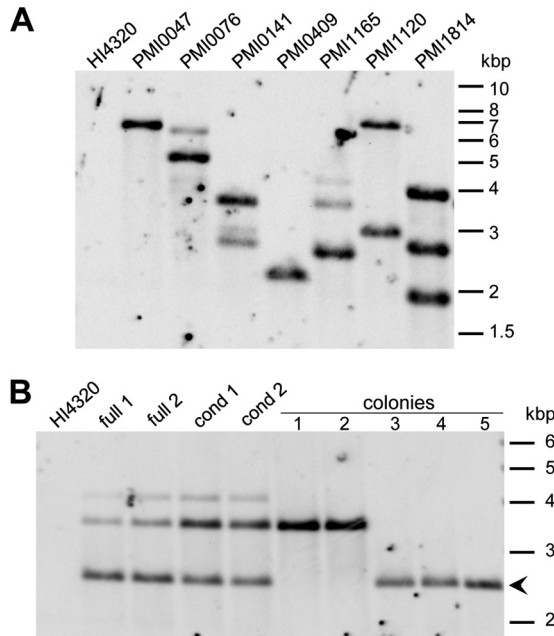

**FIG 4** Library wells with single calls often contain multiple transposons. Southern blots show digested gDNA with a probe targeting the kanamycin resistance gene in the transposon. (A) A selection of the original 25 mutants, all of which were bioinformatically called to have single insertions. gDNA was purified from the primary streak of bacteria resurrected from the library well. The predicted gene insertion location is labeled on the top. (B) The PMI1165 mutant was resurrected from both copies of the full and condensed ordered libraries, and gDNA purified from the primary streak, confirming the three-transposon pattern detected in panel A. Five single colonies from the full library were also blotted, showing isolation of expected PMI1165 insertion in three colonies (arrowhead). For both blots, wild-type HI4320 is in the first lane, and size markers are shown on right in kilobase pairs.

**Native transposases did not drive transposon hopping.** Between the earlier arbitrary PCR outcome and the possibility of new transposon insertions following colony passage (Fig. 3B, lane 7), we sought to determine whether transposon hopping was occurring in the *P. mirabilis* mutants. To investigate whether native *P. mirabilis* transposases could mediate *mariner* transposon instability, we deleted the transposase gene from pSAM_AraC, resulting in plasmid pSAMΔTnase, and repeated the mating experiment. This resulted in a small number of kanamycin-resistant colonies at approximately 1,000-fold reduced efficiency compared with pSAM_AraC. Every single colony that resulted from mating with pSAMΔTnase was also ampicillin-resistant, suggesting that these were plasmid cointegrants and not true transposon mutants. We also observed a surprising diversity in colony sizes from the control experiment using intact pSAM_AraC, including tiny colonies that formed underneath larger, normal-sized colonies. Although a native transposase might not be active during these mating conditions, taken together, the results suggest that colony contamination is very easy to obtain and transposon instability is less likely to be the source of our library issues. Furthermore, BLAST (34) of either the protein or nucleotide sequences of the transposase encoded by pSAM_AraC against the HI4320 genome did not yield any hits, so we predict native transposases encoded by HI4320 act on different lineages of mobile genetic elements.

**Construction of a condensed ordered library.** Although the full ordered library contains a wealth of mutants to pull from depending on experimental goals, it is often desirable to focus on a subset of mutations. To facilitate studies into gene function, we built a condensed ordered library consisting of 1,728 mutants that only contained predicted single transposon insertion events, with one insertion per gene. Toward this goal, we generated a list of wells containing a single predicted insertion that had the location mapped with high certainty (i.e., mapped to exactly four sequencing pools). Any gene with a certainty designation other than "unique" was hand curated to select the mutation closest to the start site with the highest confidence of plate/well identification. From this list, we picked 1,728 mutants into a new condensed ordered library, representing 45.0% of predicted genes in *P. mirabilis* HI4320,

**TABLE 2** Comparison of condensed ordered libraries

| Library feature | *P. mirabilis* | *K. pneumoniae*[a] | *E. coli*[b] |
|---|---|---|---|
| Open reading frame disrupted | 1,728 (45)[c] | 3,733 (69) | 2,913 (53) |
| Unique[d] | 1,694 (98) | 3,605 (97) | 2,851 |
| Plate-ok, unique[e] | 15 (0.9) | 46 (1) | 34 |
| Plate-ok, nonunique[f] | 0 (0) | 2 (<0.1) | 7 |
| Well-ok, unique[g] | 4 (0.2) | 38 (1) | 3 |
| Well-ok, nonunique[h] | 0 (0) | 3 (<0.1) | 3 |
| Heuristic[i] | 15 (0.9) | 21 (1) | 15 |
| Operons disrupted[j] | 480 (70) | | |

[a]Reference 11.
[b]Reference 12.
[c]Total number followed by percentage in parentheses.
[d]Insertions that were uniquely mapped to a single location in the library.
[e]Insertions for which reads with substantial counts found in only 2 plate coordinate subpools, and the highest reads for the well coordinate subpools used to map the transposon to the most probable location (the subpools with highest reads are substantially higher than the second highest subpool read count).
[f]Insertions for which reads with substantial counts found in only 2 plate coordinate subpools, and the highest reads for the well coordinate subpools used to map the transposon to the most probable location (multiple locations possible).
[g]Insertions for which reads with substantial counts found in only 2-well coordinate subpools, and the highest reads for the plate coordinate subpools used to map the transposon to the most probable location (the subpools with highest reads are substantially higher than the second highest subpool read count).
[h]Insertions for which reads with substantial counts found in only 2-well coordinate subpools, and the highest reads for the well coordinate subpools used to map the transposon to the most probable location (multiple locations possible).
[i]Insertions for which the most probable location was determined by sorting the read counts for both plate and well subsets if read counts were substantially higher than the second highest subpool read count.
[j]Predicted transcriptional units containing two or more genes.

or 49.9% of nonessential genes (Table 2 and Table S5). When more than one insertion in a particular gene was available, the one located closer to the beginning of the gene was selected. Over half of the insertions were in the first third of the coding sequence (Fig. 5A). Of 681 predicted multigene operons (35, 36), 480 (70%) are represented in the condensed library. Most mutants grew well in the condensed library (Fig. 5B); however, for any mutants that did not grow well in the condensed ordered library (i.e., optical density at 600 nm [$OD_{600}$] <0.2), mutants were repicked into a new plate and recultured for inclusion in the condensed library. Mutants in this plate likely represent transposon insertions that decrease fitness in LB static culture (Table S5, plate 219). This condensed ordered library will facilitate future experiments by allowing surveys of the largest number of mutated genes with the smallest number of mutants.

## DISCUSSION

We have generated an ordered library of *P. mirabilis* transposon mutants that will provide the opportunity to screen for virulence-related phenotypes or any *in vitro* phenotypes with a clear readout. However, we encountered significant obstacles in its preparation, in contrast with libraries our group previously constructed in *E. coli* and *K. pneumoniae* (11, 12). The major difficulty arose when we found a lower-than-expected success rate in confirming the presence of predicted mutants in specific wells of the library; that is, 3 of 25 (12.5%) randomly selected mutants could not be validated by PCR. The 87.5% validation rate and the 15% rate of multiple transposons mapped to single library wells caused us to take a closer look at the library, and subsequent issues, in particular the high rate of mixed clones, were found. However, despite these issues, the overwhelming majority of the mutants mapped correctly, and thus, this library will be useful for screening with careful experimental design, controls, and validation of all hits.

**Most library mutants need to be isolated before use.** We found that library wells with single transposon calls frequently contained additional transposons (Fig. 4A). Correspondingly, the number of wells with multiple transposons mapped (Table 1) is an underestimate. A close reading of the original ordered library technique revealed this group also identified "some frequency of picking multiple clones to the same

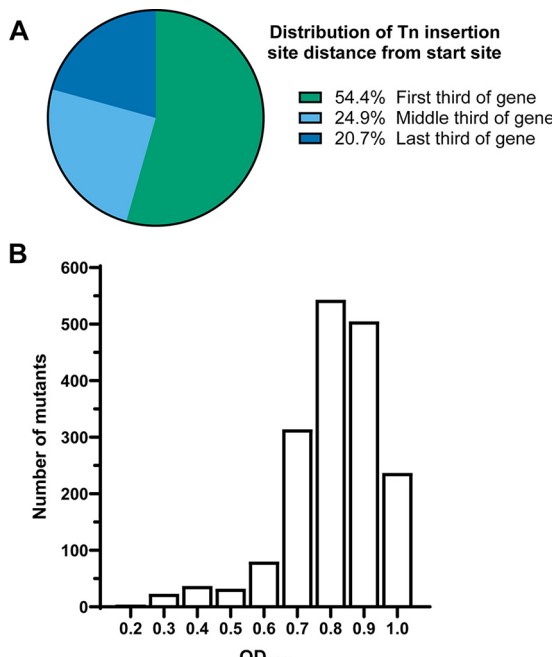

**FIG 5** Characteristics of the condensed ordered library. Mutants from the full ordered library were selected to generate a second ordered library containing one representative of each gene and the transposon insertion site as close to the start codon as possible. (A) Distribution of insertions in the condensed library. (B) Mutants within the condensed ordered library had high optical density after overnight incubation. Mutants were picked into LB medium in 96-well plates, incubated statically at 37°C for 18 h and $OD_{600}$ measured. The histogram represents the number of mutants with optical density within the range from 0.1 to 1.0 in increments of 0.1 $OD_{600}$ units.

well" due to bacterial clumping, and the authors suggested screening 5 to 10 colonies to obtain a pure clone (17). In a follow-up study, 11 to 14% of the wells of a *Mycobacterium bovis* library contained mixed clones (37). We estimate that the percentage is substantially higher in our library; based on the results in Fig. 4A, where we detected uncalled transposons in 5/7 single-called wells, we predict that the majority of single-called wells contain other uncalled mutants. Thus, if a library is intended to be used for pooled applications, the inclusion of unintentional mixed colonies should be considered. Pooling of our *P. mirabilis* library mutants for high-throughput screens, such as our similar work in *E. coli* (12), would likely be prone to high rates of false calls and might not be feasible at all. If a tight bottleneck is present in the model system that is being tested, adding an unknown number of additional mutants would be particularly problematic.

**Analysis of gaps in library coverage.** As an upper boundary for transposon library design, previous calculations estimated that we would need approximately 34,000 transposon mutants to achieve 99.99% genome coverage in *P. mirabilis* HI4320 (30, 38). We estimated that two ordered libraries would have 99.1% coverage of nonessential genes (Fig. S1). Thus, we considered explanations for gaps in coverage. We identified four possible reasons for gaps in genome coverage and consider them in the following paragraphs: (i) genes that are essential for growth in LB, (ii) repetitive sequences that could not be accurately mapped, (iii) a lack of TA insertion sites, and (iv) too few wells in the library had single, unique insertions mapped to them.

Previous Tn-seq studies in *P. mirabilis* have outlined the likely genes that are essential for growth in LB medium (29, 30). In those studies, 436 genes, comprising 11.6% of the predicted total genes encoded by *P. mirabilis* HI4320, were identified as essential. However, the proportion of essential genes does not sufficiently explain the low coverage in the ordered library.

Although there are some repetitive sequences in the *P. mirabilis* HI4320 genome, we did not expect this to be a major contributor to gaps in coverage. However, we did identify a few cases where this occurred. Because the sequencing reads used for mapping were very

short (25 bp), any repetitive sequences longer than 25 bp could not be unambiguously mapped to the genome. Because our 18,432-well library was not saturating, it is likely that even if we obtained transposon insertions in a given gene, insertions in only a subset of the TA sites in that gene would be present in the library. Furthermore, if a transposon insertion was only obtained in a repetitive region within an open reading frame, that gene would appear to be missing from the library. Indeed, we identified gaps in transposon coverage associated with known repetitive sequences. For example, *P. mirabilis* HI4320 encodes five type VI secretion system (T6SS) secreted effector operons, the first two genes of which are highly similar across the operons (i.e., *hcp* and *vgrG*) (39). A 3.2- to 3.5 kb-sized region is not represented in the library for three of these operons, while the remaining two were sufficiently different at the nucleotide level to allow alignment of transposon insertions; only the latter two are represented in the library. Notably, one of the "missing" pairs of genes (*pefAB*) has previously been mutated in *P. mirabilis* HI4320, so this operon is not essential for growth in LB (40). Three of the four copies of *zapE* from the *zap* metalloprotease region are likewise not represented. Importantly, some genes previously identified as essential in *P. mirabilis* were very likely false hits for the same reason (e.g., genes encoding T6SS or transposases) (30). However, although repetitive sequence explains some gaps in coverage, it is not a complete answer.

A third possibility for missed genes is a lack of TA insertion sites. Because the *mariner* transposon inserts at TA sites, the original CP-Cseq study using *Mycobacterium bovis* (BCG), which has a high %GC content (66%), noted gaps in coverage in some genes due to lack of insertion sites (17). A localized lack of TA insertion sites was also a possibility for libraries constructed for *E. coli* and *K. pneumoniae* (51% and 57% GC, respectively) (11, 12). This was much less likely to be a problem for *P. mirabilis* (39% GC) but perhaps led to issues with higher incidences of multiple insertions. In line with this observation, previously generated saturating transposon libraries in *P. mirabilis* HI4320 did not reveal any apparent gaps in coverage due to GC content (29, 30).

**Most gaps in the library are due to problems stemming from mating and plating.** Although the above explanations explain a small portion of gaps in genome coverage in the *P. mirabilis* libraries, they are insufficient to explain the magnitude of low coverage we observed. We therefore concluded the likely reason was the large number of wells that either contained multiple transposons or where no insertions were uniquely mapped (Fig. 2B). One major reason for wells to contain no mapped transposons was that only 70% of the sequencing reads mapped to the *P. mirabilis* HI4320 genome, which was considerably lower than the 89% of reads that mapped for our previous *E. coli* CFT073 library (12). This suggests the *P. mirabilis* library had a contaminating source of reads; perhaps the *E. coli* donor strain was still present in some quantity despite antibiotic selection. In this case, we would expect a library well to map to zero sequencing pools.

Mixed clone wells (with various proportions of each mutant) and multitransposon insertions could have caused missed calls that would have been detected with greater sequencing depth (Table 1). Adjusting the parameters for the number of reads required to map an insertion would increase detection of some of these mixed clone events. If gDNA purification was uneven across sequencing pools, this would exacerbate read depth issues. These situations would result in wells that had reads present in only one to three sequencing pools. Alternatively, the choice of random primers used in sequencing preparation could be better optimized for low %GC content, as the random primers we used followed the same method used for higher %GC organisms. However, the number of reads per insertion seems sufficient to detect insertions (Table S1).

**Recommendations for refining ordered library construction.** If we were to reconstruct this library, we would recommend taking great care to obtain sufficiently isolated colonies to reduce picking of mixed colonies. In an effort to improve conjugation efficiency, a longer mating duration was used compared with the previous use of this protocol in *P. mirabilis* (2 versus 24 h) (29, 30). This could have contributed to both the difficulty in picking single colonies and the higher-than-expected proportion of multiple transposons present in a single genome. We found tremendous variation in colony size after mating, where some colonies took more than 24 h to become easily visible and could, under a microscope, be seen to grow underneath larger colonies. This slow initial growth or small colony size did not necessarily

correlate with the robustness of the mutant after restreaking. Borgers et al. (37) found that human hands were more skilled at picking single colonies during library preparation and recommended against using a robot for picking colonies, as we used here. This issue is certainly species dependent, but it seems likely to hold true for *P. mirabilis*.

In future libraries, we would recommend (i) extensively troubleshooting mating parameters to select the best combination of single insertions and number of colonies (41); (ii) diluting the outputs (e.g., using more or larger plates [17]) to avoid picking mixed colonies, and (iii) fine-tuning postmating selection to eliminate breakthrough of the donor strain or choosing a mutagenesis method that avoids the need for a donor strain, such as transducing phage or electroporation of transposon/transposase complexes (37, 42). Longer incubation of plates will also reveal if large numbers of smaller or slower-developing colonies and/or possible donor strain breakthrough events are likely to be forming within larger, earlier-developing colonies. If transposon instability turns out to be a source of failed validation, we could consider using a different transposon. Alternatively, we could employ a modified transposon that would be more stable. For instance, ISceIM can be used to remove both the antibiotic resistance marker and sometimes one or both inverted repeats of the *mariner* transposon, making it unable to hop further (37). Future arrayed libraries could also incorporate dialable knockdown of message using technologies such as Mobile-CRISPRi, which shows promise but needs to be optimized for higher transfer efficiency in *P. mirabilis* (43).

**Proposed uses for the *P. mirabilis* ordered library.** Taken together, the unexpectedly high rates of missed calls and apparent picking of mixed colonies led to the development of an ordered library that we believe must be carefully validated before use. We propose the best use of this library would be for experiments that do not require pooling. Any assay that can be adapted to a 96-well plate format would be a good use for this library. Hits from such a screen could be independently verified for transposon insertion accuracy using the techniques like those we deployed here, and we would recommend following up with construction of clean independent mutations (24). Alternatively, for smaller-scale studies, it may be feasible to PCR verify single-colony streaks of specific wells before pooling mutants. Even though only 45% of open reading frames are represented in this library, the overwhelming majority (87.5%) of these calls have been validated by PCR. When a mutant is mixed with other clones, it can be purified by single-colony passaging.

We further note that it was possible to obtain single insertion mutants in some cases where multiple transposons were predicted in a given well of the library, as we described for *flaA* (Fig. 3B). In that instance, we preserved the purified *flaA* mutant in its own glycerol stock. Although laborious, if applied to genes that are present as mixed clones in the full library but missing from the condensed library, these extra steps would increase single gene coverage to include up to 70% of genes. The full library could also be useful for future studies targeting regions that were not selected for the condensed library, which only consists of single insertions per open reading frame. For example, RNA-seq or other experiments might identify small RNAs or other relevant features that could be present in the full library.

In summary, although the library had low overall genome coverage, in most cases when a single insertion was called, a pure culture of the desired mutant could be obtained by single-colony passaging. Most operons are represented in the library, potentially expanding its utility. However, the substantial issues with library coverage and mixed clones mean the user must exercise caution when using the library, particularly for large-scale studies. Finally, we could not completely rule out transposon instability as a source for some of the unexpected results. While the ordered library technique can be very powerful, it must be carefully executed with the proper controls. Some bacterial species, apparently, including *P. mirabilis*, will require extra vigilance in the setup of the library. Nevertheless, we anticipate that even a library with lower-than-desired coverage will be a tremendously useful tool for further studies.

## MATERIALS AND METHODS

**Bacterial strains and plasmids.** *P. mirabilis* strain HI4320 was isolated from the urine of an elderly female nursing home patient with a long-term (≥30 days) indwelling catheter (26, 44). This strain is well established as a model organism for *P. mirabilis* virulence studies and readily produces experimental UTIs in mice (23, 45, 46). *Escherichia coli* S17λpir/pSAM_AraC was used as the donor strain for mating (Addgene 91569) (30). Bacteria were routinely cultured at 37°C with aeration in LB (10 g/L tryptone, 5 g/L yeast extract, and 0.5 g/L NaCl) or on LB solidified with 1.5% agar. As needed, antibiotic selection of kanamycin (25 $\mu$g/mL), tetracycline (15 $\mu$g/mL), or ampicillin (100 $\mu$g/mL) was applied.

**Construction of a *P. mirabilis* ordered library.** A library of random transposon mutants was created by mating a fresh lawn of *P. mirabilis* HI4320 (recipient; tetracycline-resistant) with *E. coli* S17λpir/pSAM_AraC (donor, harbors the *Himar1 mariner* transposon [47]) cultured to midlogarithmic phase as previously described (41) with the following modification: during conjugation, cultures were allowed to incubate in the presence of 10 mM arabinose on 0.45-$\mu$m filter disks at 30°C for 24 h.

To verify the presence of the transposon in tetracycline-resistant, kanamycin-resistant, ampicillin-sensitive recipient colonies and that pSAM_AraC was suicidal under the mating conditions, gDNA was extracted from individual mutants (DNeasy blood and tissue kit; Qiagen) and PCR was performed using primers specific to the transposon and to the plasmid backbone as previously reported (30). Randomness of transposon insertions was confirmed by Southern blotting using a digoxigenin-labeled probe that was complementary to the kanamycin resistance cassette (41).

Following verification, 18,432 mutants were picked from LB agar containing kanamycin and tetracycline and inoculated into 192 96-well plates with LB containing kanamycin using a Qpix2 colony picker (Molecular Devices, LLC) optimized to select both large and small colony morphology. Plates were incubated statically at 37°C overnight. Any wells that appeared to be uninoculated were reinoculated by hand and plates were further incubated until turbidity was apparent. Next, a replicate of the mutant library was made by diluting the original library 1:100 in LB containing kanamycin and incubating statically at 37°C overnight. Then, 50% glycerol was added 1:1 to all wells of all plates (final concentration of 25%) before storing at −80°C.

**CP-CSeq pooling.** The genomic location of the transposon insertion in each of the mutants within the library was identified using a combination of next-generation Illumina sequencing and the CP-CSeq technique (17). Pooling of samples and preparation of fragments for sequencing were achieved as previously described (17) with modifications.

Briefly, the CP-CSeq strategy uses combinatorial pooling to reduce the total number of samples to be sequenced from 9,216 to 40 samples. This was accomplished by combining all mutants in the same row (8 samples) and all mutants in the same column (12 samples) to preserve the X,Y location within a set of 96-well plates. To preserve the Z location of each mutant, all mutants in a single plate were combined and redistributed into a separate 96-well plate. Then, the rows and columns were similarly condensed into 8 row samples and 12 column samples. In this way, each well of the library was represented in four pools (XY row, XY column, Z row, Z column) that could be bioinformatically deconvoluted postsequencing to derive the physical location of each mutant in the ordered library.

Mutants with identical genomic transposon insertion loci in one of each pool type can be unambiguously assigned to a well in the library. Mutants with transposon insertion loci identified in fewer than four pools are not given a well designation. Mutants with transposon insertion loci identified in more than four pools are assigned to a well with various degrees of certainty as explained in Vandewalle et al. (17).

To accommodate all 18,432 mutants (192 96-well plates), the plates of the ordered library were grouped into 2 sets of 96-well plates, and correspondingly, 2 sets of CP-CSeq pools were made. In addition, all steps were performed in duplicate to create a second copy of the library.

**Illumina sequencing.** Upon completion of pooling, bacteria were pelleted, supernatant was removed, and gDNA was harvested (DNeasy blood and tissue kit; Qiagen). Genomic DNA was prepared for Illumina sequencing using the Riptide targeted library prep kit (iGenomX, Twist Bioscience). Briefly, transposon-gDNA junctions were enriched using single primer PCR with homology to the transposon inverted repeat region and biotinylated nucleotides (all primer sequences can be found in Table S6). The primer binding site was a few nucleotides upstream from the end of the inverted repeat to allow for positive sequence identification in resulting sequencing reads. The resulting single-stranded, randomly sized fragments were captured with streptavidin-coated beads and gDNA washed from the sample. To release the fragments from the beads and generate double-stranded DNA, a second round of single primer PCR was performed using a primer with random nucleotides at the 5′ end and partial Illumina I7 adaptor sequence at the 3′ end. Following size selection, sequencing barcodes were added by three primer PCR using a primer with homology to the transposon and Illumina I5 adaptor and a pair of NEBNext multiplex oligonucleotides for Illumina (New England Biolabs) with eight nucleotide barcodes. Each sample was gel purified to size select for fragments ∼350 bp on a 1.5% agarose gel using GelRed stain (Millipore Sigma) to prevent UV damage during visualization. Each of the 80 samples from the subpools was assessed for purity and concentration via TapeStation (Agilent) and subjected to single end 50 NextSeq 500 sequencing. Each lane was spiked with 15% bacteriophage $\phi$X DNA to introduce diversity. Sequencing and quality control were performed at the University of Michigan Advanced Genomics Core. Raw reads are available through NIH BioProject accession number PRJNA608758 and barcodes are provided in Table S7. Each read was trimmed and transposon insertion location mapped to the *P. mirabilis* HI4320 genome or its single plasmid (28). Deconvolution of location within the ordered library was determined using CLC Genomics Workbench and open-source platform Galaxy workflow described in Vandewalle et al. (17) (https://galaxyproject.org/) by the University of Michigan Bioinformatics Core. To match the library location with transposon insertion location and gene coordinates, the Fuzzy Join function of the Fuzzy Lookup Add-In for Microsoft Excel was used. Sequencing pool details are shown in Table S1.

**Calculation of predicted genome saturation.** Prediction of transposon saturation of genes in the ordered library was calculated using the "coupon collector's problem" as previously used by Vandewalle et al. to estimate library saturation for *M. bovis* (17, 48). Specifically, of 329,520 TA sites in the *P. mirabilis* HI4320 genome, 296,433 were calculated to be within nonessential genomic regions, as defined by previous Tn-seq screens (29, 30). Based on this, we calculated the average distance between TA transposon insertion sites to be approximately every 12.4 bp in *P. mirabilis* HI4320. Using an average gene size of 941 bp, the median number of TA sites per gene is 76. The portion of the genome encoding nonessential genes was then cut into 3,907 nonessential genome regions ("genes"), each containing 76 TAs. This resulted in an estimation of 34,561 mutants to attain 100% saturation. The probability of saturation for any given number of transposon mutants was then calculated with assistance from the Consulting for Statistics, Computing and Analytics Research (CSCAR) resource at the University of Michigan.

**Validation of ordered library well assignments.** To confirm mapping of mutants to the ordered library, we aimed to pick a mutant from every 7 to 8 plates in the library (192/25); this would ensure that plates across the library were well represented. We then sorted the wells with single insertions by genomic location, with the chromosome and plasmid concatenated. Scrolling from the origin to the terminus of the genome, the next mutant that matched to the desired plate number was selected. This method was effective at ensuring diverse genomic and library coverage for the selection of 25 random mutants.

The contents of 25 random wells in the library were PCR-verified using transposon-specific primer CP7 paired with gene-specific primers (Table S6). For wells that did not verify, arbitrary PCR was used as previously described (31, 32). Briefly, a set of nested PCR primers was used to amplify *P. mirabilis* sequence adjacent to the transposon insertion site. The resulting PCR product was then cloned into pCR4-TOPO (Invitrogen), electroporated into *E. coli* Top 10, and sent for Sanger sequencing. Sequences containing transposon-chromosome junctions were aligned with *P. mirabilis* HI4320 to identify the site of transposon insertion.

**Southern blot.** Genomic DNA was purified using the DNeasy blood and tissue kit (Qiagen). For most applications, a single colony of bacteria was cultured in LB medium. Otherwise, for Southern blots where the contents of a library well were being assayed, a loopful of the primary streak of bacteria on LB agar was added to 500 $\mu$L of water and collected by centrifugation. To track transposon insertions, two Southern blot methodologies were used. The initial blot to confirm random transposon insertions was conducted as previously described using DNA digested with HindIII (41). For all subsequent blots, gDNA was digested overnight with HindIII-HF and SspI-HF. A digoxigenin-labeled DNA probe complementary to the kanamycin resistance gene was used to detect fragments containing the transposon (Table S6). Southern blot hybridization was conducted using the DIG-High Prime DNA Labeling and Detection kit (Millipore Sigma) following the manufacturer's instructions.

**Swarming motility.** Swarming motility was assessed as previously described (33). Briefly, 5 $\mu$L of an overnight culture of *P. mirabilis* in LB medium was spotted onto swarm agar (LB with 10 g/L NaCl), allowed to dry, and incubated at 30°C overnight. Swarming was recorded as surface migration of bacteria from the point of inoculation.

**Transposase gene deletion.** The Gibson method (49) was used to delete the gene encoding the transposase and its promoter from pBAD_AraC. Briefly, oligonucleotide primers with overlapping sequences were designed to PCR-amplify pSAM_AraC minus the 1,445 bp containing the pBAD promoter and the transposase gene (Table S6). Gibson Assembly master mix (New England Biolabs) was used to reconstruct the truncated plasmid, which was then introduced into *E. coli* S17$\lambda$pir by electroporation. Ampicillin-resistant colonies were screened for deletion using PCR. The resulting plasmid, pSAMΔTnase, was used in mating experiments with *P. mirabilis* HI4320 as described above.

**Condensed ordered library.** To create a *P. mirabilis* library with the single best representative mutant for each gene possible, the full ordered library was condensed as previously described for *Klebsiella pneumoniae* (11), taking special care to sort out instances where more than one insertion was identified in a single well. Each mutant selected for the condensed library was resurrected from the full ordered library and rearrayed by inoculating LB and incubating statically at 37°C overnight. Following incubation, OD$_{600}$ in each plate was measured, glycerol was added (final concentration of 25%), and the library was stored at −80°C. Any wells that did not reach an OD$_{600}$ of 0.2 were inoculated into a new plate with LB containing kanamycin for a second attempt at recovery for the mutant. These insertions likely interrupt genes with sublethal growth defects in LB.

**Data availability.** Raw Illumina sequencing reads of CP-CSeq pools are available through NIH BioProject at accession number PRJNA608758.

## SUPPLEMENTAL MATERIAL

Supplemental material is available online only.
**SUPPLEMENTAL FILE 1**, PDF file, 0.7 MB.
**SUPPLEMENTAL FILE 2**, XLSX file, 1.1 MB.

## ACKNOWLEDGMENTS

Research reported in this publication was supported by National Institutes of Health award R01AI059722 (to H.L.T.M and M.M.P.).

We gratefully acknowledge Weisheng Wu at the University of Michigan Bioinformatics Core for bioinformatics contributions, Kristof Vandewalle for interpretation of sequencing

pool mapping, Corey Powell of Consulting for Statistics, Computing and Analytics Research (CSCAR) at U. Michigan for assistance with calculating theoretical library saturation, and Madison Fitzgerald for editing the manuscript. Fig. 3A was created using BioRender.

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
