## [Reviewer comments · Microbiology Spectrum]

Microbiology Spectrum

Construction of an ordered transposon library for uropathogenic *Proteus mirabilis* HI4320

Melanie Pearson, Sapna Pahil, Valerie Forsyth, Allyson Shea, and Harry Mobley

Corresponding Author(s): Melanie Pearson, University of Michigan Medical School

Review Timeline:

Submission Date:	August 10, 2022
Editorial Decision:	September 17, 2022
Revision Received:	October 25, 2022
Accepted:	November 1, 2022

Editor: Philip Rather

Reviewer(s): The reviewers have opted to remain anonymous.

Transaction Report:

DOI: <https://doi.org/10.1128/spectrum.03142-22>

September 17, 2022

Dr. Melanie M Pearson
University of Michigan Medical School
Microbiology and Immunology
5641 Medical Science Bldg II
1150 W Medical Center Dr
Ann Arbor, MI 48109-0620

Re: Spectrum03142-22 (Construction of an ordered transposon library for uropathogenic *Proteus mirabilis* HI4320)

Dear Dr. Pearson,

Thank you for submitting your manuscript to Microbiology Spectrum. Your manuscript has been reviewed by two experts in the field. Both reviewers found merit to your study, but have a large number of concerns that need to be carefully addressed. When submitting the revised version of your paper, please provide (1) point-by-point responses to the issues raised by the reviewers as file type "Response to Reviewers," not in your cover letter, and (2) a PDF file that indicates the changes from the original submission (by highlighting or underlining the changes) as file type "Marked Up Manuscript - For Review Only". Please use this link to submit your revised manuscript - we strongly recommend that you submit your paper within the next 60 days or reach out to me. Detailed instructions on submitting your revised paper are below.

Link Not Available

Sincerely,

Philip Rather

Journals Department
Reviewer comments:

Reviewer #1 (Comments for the Author):

The manuscript describes an arrayed transposon mutant library with low-medium genome coverage for *Proteus mirabilis*, an important pathogen, using a pooling strategy to identify mutants in individual wells of the array. Such libraries are valuable resources for the research community for carrying out functional genomic studies. Unfortunately, it seems the *P. mirabilis* library construction was beset by technical difficulties due to cross-contamination of mutants and low sequence depth in the analysis that seriously compromise its utility. However, suggestions for accommodating them should help make the library useable. The manuscript was challenging to follow, and some key information was absent, and these should be addressed for publication.

1. Missing sequencing metrics table. A glaring omission is the lack of a sequencing metrics table giving the reads/Illumina run, the fraction mapped successfully, number of insertions represented in each run, and average reads/insertion in each run. An explanation for some of the technical difficulties found in the library construction would be low sequence coverage, and sequence metrics are fundamental to the study and certainly need to be included.
2. Table S1. Gene names as well as loci need to be provided.
3. line 78- A summary of the CP-Cseq pooling procedure is needed so that the reader doesn't have to refer to the detailed Methods section and earlier paper to understand it.
4. Fig 2B- Although all the wells had mutants, 42.5% could not be assigned reads. No explanation is provided for this difficulty, and something should be said about the low efficiency of mutant detection. Perhaps the most likely explanation is low sequence coverage? Sequencing metrics would be useful for evaluating.
5. line 103/Table 2- It's worth mentioning that the number of wells with multiple transposon mutants in them must be even higher than estimated due to the apparent low efficiency of detection referred to in point 4.
6. Fig. S2. The validation study showing 22/25 wells contained predicted mutants by PCR was a strong result. Of course, the result does not rule out the possibility of additional undetected mutations in the same wells, and this should be stated. Indeed, studies described later found that 5/7 wells showed multiple mutations (Fig. 4A).
7. It's disturbing that of the three wells not containing predicted mutants (point 6.), only one could be further analyzed, but it was found to have two different unpredicted mutations. This indicates that many of the wells thought to contain single mutations likely carry undetected mutations.
8. Line 77-80. I don't understand how the original *flaA* mutant was motile. Contamination with a second motile strain?
9. line 205- given the extensive cross-contamination of wells, it's too bad the mutants in the condensed library weren't single colony purified. Purification and re-sequencing would have made this a much more useful resource.
10. Will strains we distributed to the research community? If so, instructions for obtaining them should be included.

Reviewer #2 (Comments for the Author):

Review of Pearson et al., "Construction of an ordered transposon library for uropathogenic *Proteus mirabilis* HI4320"

Overview:

The authors use a new methodology to produce a structured transposon library in a model organism for urinary tract infections (a first example in *Proteus*?). This manuscript provides data supporting the library and discusses the gains/pitfalls of the library construction. The library could help initial mutagenesis screens for field-specific researchers. For the reader, most informative are the challenges and suggestions for future library constructions using these methods. However, it was difficult to parse this manuscript, and the most helpful information is missing emphasis and details. I've separated comments into major and minor below.

Major:

I strongly encourage the authors to state what they think is the cause of the library problems more explicitly. Start with the condensed library as a solution and then explain their rationale using the challenges as examples. The current manuscript ends with a final product (the condensed, structured transposon library), but most of the data speak to challenges in the initial version. Early statements in the text that seems like facts (e.g., lines 88 - 91) reflect the "draft" library. The reader can come away feeling that the statistics for "draft" (or initial) library (with errors) are the final ones. Then, the last portion of the manuscript reveals the final (usable to others) product and that the draft library is not usable. As stated above, I recommend that the authors focus first on the final library and its statistics and then do a retrospective framing of the challenges and caveats of this process. Here are some detailed suggestions:

- From this reader's perspective, the heart of the issue with the draft library seems to be how the initial colonies were chosen. The presented data points to a mistake in the initial arraying of the library, leading to some wells having mixed clones with varying fitness. Potential causes include the following:
 - o The colonies were chosen with a colony picker rather than by hand. See lines 230 - 242: The study by Vandewalle et. al., 2015, used hand-picked colonies and showed no signs of mixed clones in their validation process. The follow-up study referred to a new library they did. They mentioned that picking five to ten clones from the glycerol stocks of the library was sufficient to recover the mutant in a clonally pure form. This step is a common good microbiology practice as some frequency of picking

multiple clones to the same well is unavoidable during ordered library preparation (due to strong clumping behavior). For lines 240 - 242, the comparison should reflect manual or robot picking. Discuss the technical methods earlier in the manuscript.

o Perhaps there was some swarming on the initial plates. The authors used 1.5% LB agar, which seemed to allow for partial swarming in their hands (Figure S3B) and was reported to support swarming most recently in [Little K, Austerman J, Zheng J, Gibbs KA. Cell Shape and Population Migration Are Distinct Steps of *Proteus mirabilis* Swarming That Are Decoupled on High-Percentage Agar. *J Bacteriol.* 2019 May 8;201(11):e00726-18. doi: 10.1128/JB.00726-18. PMID: 30858303; PMCID: PMC6509654.]

• Regarding construction of the ordered library:

o Unclear is the basis for the model for determining the percent coverage of the transposon mutagenesis methodology. Please add clarifying comments to lines 76 - 85 or the methods.

What were the parameters for the 'coupon collector' model? Will there be differences between libraries? Is it based on TA sites in the genome?

The model predicts that with 18,432 mutants, 99.1% of the non-essential genes should be hit, which is above what the data set shows. Can you address this issue in the text? You assume that doubling the number of mutants will allow disrupting almost all targetable genes in this genome, yet this is not the case. Please discuss.

The TA sites do not seem to be a likely cause of the draft library problems. See the above discussion of the arraying.

• Figure 1 is for the draft library, but the reader subsequently learns that a portion of the wells is incorrect. The authors do not provide an equivalent image for the final ("condensed") library; table S3 is insufficient. Instead, consider moving Figure 1 to the supplemental figures.

• Consider lines 132 - 142; this passage undercuts the entire library, raising questions as to why the draft library was discussed up until now.

o Also, the other two "new" mutations are in PMI_RS18625 (DNA topoisomerase) and PMI_RS00680 (RecR). The authors should include this information instead of requiring readers to find the information. Do the authors suspect certain gene regions are susceptible to incorrect insertions? Is there unexpected homology?

• Much of the text in the discussion considers the draft library; however, that tool is not usable to the research community. Please consider re-writing the discussion to mainly focus on the condensed library, as that is the product supporting this manuscript's publication.

Some relevant references are missing:

(a) The authors cite transposon libraries in *P. mirabilis* strain HI4320. Yet, several groups published mutagenesis libraries in other strains (see below). A robust consideration of the challenges/constraints in those libraries is important as a justification in the introduction. Potential references are:

• Belas R, Erskine D, Flaherty D. Transposon mutagenesis in *Proteus mirabilis*. *J Bacteriol.* 1991 Oct;173(19):6289-93. doi: 10.1128/jb.173.19.6289-6293.1991. PMID: 1655704; PMCID: PMC208382.

• Stevenson LG, Rather PN. A novel gene involved in regulating the flagellar gene cascade in *Proteus mirabilis*. *J Bacteriol.* 2006 Nov;188(22):7830-9. doi: 10.1128/JB.00979-06. Epub 2006 Sep 15. PMID: 16980463; PMCID: PMC1636314.

• Gibbs KA, Urbanowski ML, Greenberg EP. Genetic determinants of self identity and social recognition in bacteria. *Science.* 2008 Jul 11;321(5886):256-9. doi: 10.1126/science.1160033. PMID: 18621670; PMCID: PMC2567286.

• Peters JM, Koo BM, Patino R, Heussler GE, Hearne CC, Qu J, Inclan YF, Hawkins JS, Lu CHS, Silvis MR, Harden MM, Osadnik H, Peters JE, Engel JN, Dutton RJ, Grossman AD, Gross CA, Rosenberg OS. Enabling genetic analysis of diverse bacteria with Mobile-CRISPRi. *Nat Microbiol.* 2019 Feb;4(2):244-250. doi: 10.1038/s41564-018-0327-z. Epub 2019 Jan 7. PMID: 30617347; PMCID: PMC6424567.

(b) References and consideration of equivalent, genome-scale deletion libraries are missing (lines 39 - 41). Discussing these references would help this paragraph. Missing is the broad scope and something about the limitations or other disadvantages this study tries to overcome (cost, time-consuming?). Potential references are:

• Koo, B.-M. et al. Construction and analysis of two genome-scale deletion libraries for *Bacillus subtilis*. *Cell Syst.* 4, 291-305.e297 (2017).

• Baym, M., Shaket, L., Anzai, I. A., Adesina, O. & Barstow, B. Rapid construction of a whole-genome transposon insertion collection for *Shewanella oneidensis* by Knockout Sudoku. *Nat. Commun.* 7, 13270 (2016).

• Porwollik, S. et al. Defined single-gene and multi-gene deletion mutant collections in *Salmonella enterica* sv Typhimurium. *PLoS ONE* 9, e99820 (2014).

The consideration of the bottleneck effect is confusing as written. This limitation appears due to all transposon libraries, not just those constructed with CP-Cseq. If retaining, consider expanding on the term "bottleneck issue" and put in the context of the larger field. How did your study address this limitation? How is it different (better?) than other approaches (consider the above references for additional approaches)? Please directly address these concerns or reframe the "bottleneck effect" limitations in the revised text, both in the introduction and discussion. Lines 241 and 242 should be edited to clarify the authors' points.

The paragraph starting at line 122 would benefit from more clarity. For example:

• Line 122: "Anchored PCR"? Please clarify the terminology, as this technique is standard.

• Line 125: explicitly state that the insertion was confirmed.

• Lines 129 - 131 reference "unexpected" results from libraries, but the "untraceable reads" and "many wells predicted to contain multiple transposon events." Yet, the text and the numbers presented do not match the urgency of this language or reveal the

assays. Just earlier in the paragraph, the authors state that "the vast majority of the mutants [they] tested were validated" (emphasis is mine) but that the 3/25 was enough to trigger a broad introspection. These statements (and the distinction in urgency and concern) appear contradictory.

- Separately, the condensed library is a fraction of the draft library. The authors end with a cautionary note about using the condensed library as well. So, is it really the "vast majority" of mutant strains that are valid to use?

Lines 143 - 163 regarding native transposases and transposon stability:

- The presented data supports mixed clones (particularly given the colonies in Figure 3). It is unclear why the first discussed approach is to look at potential transposon hopping. The authors should consider moving the discussion of native transposases and transposon stability later or relegating it to the supplemental information.
- It seems that if native transposases are the cause, this would occur after library construction. Did the authors look at the stability of the condensed library strains, such as those in Figure 4? When the colonies were repeatedly *freshly* taken from the frozen library, did strains from the condensed library retain the mapped insertion site? If so, please report this data.
- Lines 148 - 150 argue for native transposases acting on the mariner transposon, which would be quite different from other bacteria for which mariner is used. Please include an alignment of 45 transposases with mariner transposases (either sequence or predicted structures) to support this assertion if retained.
- In addition, a scar is not guaranteed (as acknowledged in line 149 by the authors). [Note: clarify that "scar" refers to nucleotide traces of the insertion element.] As such, a single site is not sufficient to prove or disprove the hypothesis. If the authors choose to keep this section in a revised manuscript, additional mutant strains (that have new insertion sites) should also be sequenced.
- Please clarify (regarding line 145) how many transposases are in the genome of the other two referenced libraries.
- Please state the number of colonies (line 156).

Some statements are too broad, given the presented data. For example (not comprehensive):

- There were saturation differences between the chromosome and plasmid; however, the text stated that coverage was similar. Please edit lines 88 - 91 to reflect.
- Lines 120 - 121: "This method was effective at ensuring diverse genomic and library coverage." Yet, the following several sections speak to the many inaccuracies in the library, causing the formation of a "condensed" library. The condensed library has significantly less coverage in the plasmid and is not "diverse" in coverage (see Table S3).
- The statement in line 162 ("colony contamination is very easy to obtain") is overly broad and applicable to every bacterium.

Data availability:

- Please update Table S3 (condensed library) to better match equivalent studies, such as the detailed supplementary in [Vandewalle K, Festjens N, Plets E, Vuylsteke M, Saeys Y, Callewaert N. Characterization of genome-wide ordered sequence-tagged Mycobacterium mutant libraries by Cartesian Pooling-Coordinate Sequencing. Nat Commun. 2015 May 11;6:7106. doi: 10.1038/ncomms8106. PMID: 25960123; PMCID: PMC4432585].
- Minimally, include all transposon insertion events in the condensed library.

Minor

The manuscript frames comparisons to *K. pneumoniae* and *E. coli*, but the rationale for this process is unclear, especially since that data appears to be separately published.

Consider adding header sentences.

Present the data as percentages. For example, X% of the transposon insertion event happened within ORF (Figure 2C). How many genes remain unhit?

Figure 3 and lines 182 - 184, what is the band in lane 7 that is absent from the other lanes?

Staff Comments:

Preparing Revision Guidelines

- Point-by-point responses to the issues raised by the reviewers in a file named "Response to Reviewers," NOT IN YOUR COVER LETTER.

- Upload a compare copy of the manuscript (without figures) as a "Marked-Up Manuscript" file.
- Each figure must be uploaded as a separate file, and any multipanel figures must be assembled into one file.
- Manuscript: A .DOC version of the revised manuscript
- Figures: Editable, high-resolution, individual figure files are required at revision, TIFF or EPS files are preferred

Please return the manuscript within 60 days; if you cannot complete the modification within this time period, please contact me. If you do not wish to modify the manuscript and prefer to submit it to another journal, please notify me of your decision immediately so that the manuscript may be formally withdrawn from consideration by Microbiology Spectrum.

Review of Pearson et al., “Construction of an ordered transposon library for uropathogenic *Proteus mirabilis* HI4320”

Overview:

The authors use a new methodology to produce a structured transposon library in a model organism for urinary tract infections (a first example in *Proteus*?). This manuscript provides data supporting the library and discusses the gains/pitfalls of the library construction. The library could help initial mutagenesis screens for field-specific researchers. For the reader, most informative are the challenges and suggestions for future library constructions using these methods. However, it was difficult to parse this manuscript, and the most helpful information is missing emphasis and details. I’ve separated comments into major and minor below.

Major:

I strongly encourage the authors to state what they think is the cause of the library problems more explicitly. Start with the condensed library as a solution and then explain their rationale using the challenges as examples. The current manuscript ends with a final product (the condensed, structured transposon library), but most of the data speak to challenges in the initial version. Early statements in the text that seems like facts (e.g., lines 88 – 91) reflect the “draft” library. The reader can come away feeling that the statistics for “draft” (or initial) library (with errors) are the final ones. Then, the last portion of the manuscript reveals the final (usable to others) product and that the draft library is not usable. As stated above, I recommend that the authors focus first on the final library and its statistics and then do a retrospective framing of the challenges and caveats of this process. Here are some detailed suggestions:

- From this reader’s perspective, the heart of the issue with the draft library seems to be how the initial colonies were chosen. The presented data points to a mistake in the initial arraying of the library, leading to some wells having mixed clones with varying fitness. Potential causes include the following:
 - The colonies were chosen with a colony picker rather than by hand. See lines 230 – 242: The study by Vandewalle et. al., 2015, used hand-picked colonies and showed no signs of mixed clones in their validation process. The follow-up study referred to a new library they did. They mentioned that picking five to ten clones from the glycerol stocks of the library was sufficient to recover the mutant in a clonally pure form. This step is a common good microbiology practice as some frequency of picking multiple clones to the same well is unavoidable during ordered library preparation (due to strong clumping behavior). For lines 240 – 242, the comparison should reflect **manual** or **robot** picking. Discuss the technical methods earlier in the manuscript.
 - Perhaps there was some swarming on the initial plates. The authors used 1.5% LB agar, which seemed to allow for partial swarming in their hands (Figure S3B) and was reported to support swarming most recently in [Little K, Austerman J, Zheng J, Gibbs KA. Cell Shape and Population Migration Are Distinct Steps of *Proteus mirabilis* Swarming That Are Decoupled on High-Percentage Agar. J Bacteriol.

2019 May 8;201(11):e00726-18. doi: 10.1128/JB.00726-18. PMID: 30858303; PMCID: PMC6509654.]

- Regarding construction of the ordered library:
 - Unclear is the basis for the model for determining the percent coverage of the transposon mutagenesis methodology. Please add clarifying comments to lines 76 – 85 or the methods.
 - What were the parameters for the ‘coupon collector’ model? Will there be differences between libraries? Is it based on TA sites in the genome?
 - The model predicts that with 18,432 mutants, 99.1% of the non-essential genes should be hit, which is above what the data set shows. Can you address this issue in the text? You assume that doubling the number of mutants will allow disrupting almost all targetable genes in this genome, yet this is not the case. Please discuss.
 - The TA sites do not seem to be a likely cause of the draft library problems. See the above discussion of the arraying.
- Figure 1 is for the draft library, but the reader subsequently learns that a portion of the wells is incorrect. The authors do not provide an equivalent image for the final (“condensed”) library; table S3 is insufficient. Instead, consider moving Figure 1 to the supplemental figures.
- Consider lines 132 – 142; this passage undercuts the entire library, raising questions as to why the draft library was discussed up until now.
 - Also, the other two “new” mutations are in PMI_RS18625 (DNA topoisomerase) and PMI_RS00680 (RecR). The authors should include this information instead of requiring readers to find the information. Do the authors suspect certain gene regions are susceptible to incorrect insertions? Is there unexpected homology?
- Much of the text in the discussion considers the draft library; however, that tool is not usable to the research community. Please consider re-writing the discussion to mainly focus on the condensed library, as that is the product supporting this manuscript’s publication.

Some relevant references are missing:

- (a) The authors cite transposon libraries in *P. mirabilis* strain HI4320. Yet, several groups published mutagenesis libraries in other strains (see below). A robust consideration of the challenges/constraints in those libraries is important as a justification in the introduction.

Potential references are:

- Belas R, Erskine D, Flaherty D. Transposon mutagenesis in *Proteus mirabilis*. J Bacteriol. 1991 Oct;173(19):6289-93. doi: 10.1128/jb.173.19.6289-6293.1991. PMID: 1655704; PMCID: PMC208382.
- Stevenson LG, Rather PN. A novel gene involved in regulating the flagellar gene cascade in *Proteus mirabilis*. J Bacteriol. 2006 Nov;188(22):7830-9. doi: 10.1128/JB.00979-06. Epub 2006 Sep 15. PMID: 16980463; PMCID: PMC1636314.
- Gibbs KA, Urbanowski ML, Greenberg EP. Genetic determinants of self identity and social recognition in bacteria. Science. 2008 Jul 11;321(5886):256-9. doi: 10.1126/science.1160033. PMID: 18621670; PMCID: PMC2567286.
- Peters JM, Koo BM, Patino R, Heussler GE, Hearne CC, Qu J, Inclan YF, Hawkins JS, Lu CHS, Silvis MR, Harden MM, Osadnik H, Peters JE, Engel JN, Dutton RJ,

Grossman AD, Gross CA, Rosenberg OS. Enabling genetic analysis of diverse bacteria with Mobile-CRISPRi. *Nat Microbiol.* 2019 Feb;4(2):244-250. doi: 10.1038/s41564-018-0327-z. Epub 2019 Jan 7. PMID: 30617347; PMCID: PMC6424567.

(b) References and consideration of equivalent, genome-scale deletion libraries are missing (lines 39 – 41). Discussing these references would help this paragraph. Missing is the broad scope and something about the limitations or other disadvantages this study tries to overcome (cost, time-consuming?). Potential references are:

- Koo, B.-M. et al. Construction and analysis of two genome-scale deletion libraries for *Bacillus subtilis*. *Cell Syst.* **4**, 291–305.e297 (2017).
- Baym, M., Shaket, L., Anzai, I. A., Adesina, O. & Barstow, B. Rapid construction of a whole-genome transposon insertion collection for *Shewanella oneidensis* by Knockout Sudoku. *Nat. Commun.* **7**, 13270 (2016).
- Porwollik, S. et al. Defined single-gene and multi-gene deletion mutant collections in *Salmonella enterica* sv Typhimurium. *PLoS ONE* **9**, e99820 (2014).

The consideration of the bottleneck effect is confusing as written. This limitation appears due to **all** transposon libraries, not just those constructed with CP-Cseq. If retaining, consider expanding on the term “bottleneck issue” and put in the context of the larger field. How did your study address this limitation? How is it different (better?) than other approaches (consider the above references for additional approaches)? Please directly address these concerns or reframe the “bottleneck effect” limitations in the revised text, both in the introduction and discussion. Lines 241 and 242 should be edited to clarify the authors’ points.

The paragraph starting at line 122 would benefit from more clarity. For example:

- Line 122: “Anchored PCR”? Please clarify the terminology, as this technique is standard.
- Line 125: explicitly state that the insertion was confirmed.
- Lines 129 – 131 reference “unexpected” results from libraries, but the “untraceable reads” and “many wells predicted to contain multiple transposon events.” Yet, the text and the numbers presented do not match the urgency of this language or reveal the assays. Just earlier in the paragraph, the authors state that “the **vast** majority of the mutants [they] tested were validated” (emphasis is mine) but that the 3/25 was enough to trigger a broad introspection. These statements (and the distinction in urgency and concern) appear contradictory.
- Separately, the condensed library is a fraction of the draft library. The authors end with a cautionary note about using the condensed library as well. So, is it really the “vast majority” of mutant strains that are valid to use?

Lines 143 – 163 regarding native transposases and transposon stability:

- The presented data supports mixed clones (particularly given the colonies in Figure 3). It is unclear why the first discussed approach is to look at potential transposon hopping. The authors should consider moving the discussion of native transposases and transposon stability later or relegating it to the supplemental information.
- It seems that **if** native transposases are the cause, this would occur **after** library construction. Did the authors look at the stability of the condensed library strains, such as

those in Figure 4? When the colonies were repeatedly *freshly* taken from the frozen library, did strains from the condensed library retain the mapped insertion site? If so, please report this data.

- Lines 148 – 150 argue for native transposases acting on the mariner transposon, which would be quite different from other bacteria for which mariner is used. Please include an alignment of 45 transposases with mariner transposases (either sequence or predicted structures) to support this assertion if retained.
- In addition, a scar is not guaranteed (as acknowledged in line 149 by the authors). [Note: clarify that “scar” refers to nucleotide traces of the insertion element.] As such, a single site is not sufficient to prove or disprove the hypothesis. If the authors choose to keep this section in a revised manuscript, additional mutant strains (that have new insertion sites) should also be sequenced.
- Please clarify (regarding line 145) how many transposases are in the genome of the other two referenced libraries.
- Please state the number of colonies (line 156).

Some statements are too broad, given the presented data. For example (not comprehensive):

- There were saturation differences between the chromosome and plasmid; however, the text stated that coverage was similar. Please edit lines 88 – 91 to reflect.
- Lines 120 – 121: “This method was effective at ensuring diverse genomic and library coverage.” Yet, the following several sections speak to the many inaccuracies in the library, causing the formation of a “condensed” library. The condensed library has significantly less coverage in the plasmid and is not “diverse” in coverage (see Table S3).
- The statement in line 162 (“colony contamination is very easy to obtain”) is overly broad and applicable to every bacterium.

Data availability:

- Please update Table S3 (condensed library) to better match equivalent studies, such as the detailed supplementary in [Vandewalle K, Festjens N, Plets E, Vuylsteke M, Saeys Y, Callewaert N. Characterization of genome-wide ordered sequence-tagged Mycobacterium mutant libraries by Cartesian Pooling-Coordinate Sequencing. Nat Commun. 2015 May 11;6:7106. doi: 10.1038/ncomms8106. PMID: 25960123; PMCID: PMC4432585].
- Minimally, include all transposon insertion events in the condensed library.

Minor

The manuscript frames comparisons to *K. pneumoniae* and *E. coli*, but the rationale for this process is unclear, especially since that data appears to be separately published.

Consider adding header sentences.

Present the data as percentages. For example, X% of the transposon insertion event happened within ORF (Figure 2C). How many genes remain unhit?

Figure 3 and lines 182 – 184, what is the band in lane 7 that is absent from the other lanes?

Responses to reviewers:

We received comments from two reviewers. We echo the reviewers' frustration that our *P. mirabilis* ordered library, using a technique that had been successfully carried out by our group for two other species, contained flaws that limit its utility. We worked to understand where things went wrong in the hopes that this would help other groups. At the same time, mutant construction in *P. mirabilis* remains laborious. Creation of an 18,432 mutant arrayed ordered library for this species and a condensed library that contains a single mapped mutation in 45% of predicted genes is a notable achievement. We believe the libraries, both full and condensed, will represent a useful resource when employed with careful controls. Specific responses, in blue lettering, follow below:

Reviewer #1 (Comments for the Author):

The manuscript describes an arrayed transposon mutant library with low-medium genome coverage for *Proteus mirabilis*, an important pathogen, using a pooling strategy to identify mutants in individual wells of the array. Such libraries are valuable resources for the research community for carrying out functional genomic studies. Unfortunately, it seems the *P. mirabilis* library construction was beset by technical difficulties due to cross-contamination of mutants and low sequence depth in the analysis that seriously compromise its utility. However, suggestions for accommodating them should help make the library useable. The manuscript was challenging to follow, and some key information was absent, and these should be addressed for publication.

1. Missing sequencing metrics table. A glaring omission is the lack of a sequencing metrics table giving the reads/Illumina run, the fraction mapped successfully, number of insertions represented in each run, and average reads/insertion in each run. An explanation for some of the technical difficulties found in the library construction would be low sequence coverage, and sequence metrics are fundamental to the study and certainly need to be included.

Sequencing metrics have been added as new Supplemental Table S1. Most metrics were well within expected parameters. However, the percentage of mapped reads was lower than expected (average of 70%; Table S1), and this could reflect the presence of a contaminant, such as the *E. coli* donor strain used for mating (lines 308-312). The average number of reads per insertion across all pools was 516.8 and the lowest coverage pool had an average of 255.4 reads per insertion. This should have been sufficient sequencing depth to detect insertions.

2. Table S1. Gene names as well as loci need to be provided.

We have added gene names and annotations to Table S1 (now Table S3). The loci, both gene locus and precise nucleotide where transposon insertion occurred, were in the original table.

3. line 78- A summary of the CP-Cseq pooling procedure is needed so that the reader doesn't have to refer to the detailed Methods section and earlier paper to understand it.

A summary has been added here (lines 86-93). The description of CP-CSeq in the Methods has been rewritten to be easier to follow (lines 410-424).

4. Fig 2B- Although all the wells had mutants, 42.5% could not be assigned reads. No explanation is provided for this difficulty, and something should be said about the low efficiency of mutant detection.

Perhaps the most likely explanation is low sequence coverage? Sequencing metrics would be useful for evaluating.

Sequencing metrics have been added as new Supplemental Table S1.

As noted in new Table S1 and in our reply to comment 1, 70% of reads were mapped to the *P. mirabilis* chromosome. The rest were filtered out. Thus, we believe library wells with no reads are due to breakthrough *E. coli* donor from the transposon mating (Discussion, lines 308-312). We considered exploring this further, but 1) it would not change the major conclusions and 2) bioinformatic re-analysis to determine the percentage of reads with homology to *E. coli* S17 was cost prohibitive.

Additional reasons for wells without assignments could include: 1) Poor genomic DNA extraction from pools, 2) uneven PCR amplification during sequencing fragment preparation, 3) a need to optimize fragment amplification for a low-%GC organism such as *P. mirabilis*, or 4) low quality of fragments sent for sequencing, which would result in variability of reads across pools. Some of these considerations are briefly discussed on lines 314-322.

5. line 103/Table 2- It's worth mentioning that the number of wells with multiple transposon mutants in them must be even higher than estimated due to the apparent low efficiency of detection referred to in point 4.

A new sentence in the Discussion now directly states this (lines 253-254).

6. Fig. S2. The validation study showing 22/25 wells contained predicted mutants by PCR was a strong result. Of course, the result does not rule out the possibility of additional undetected mutations in the same wells, and this should be stated. Indeed, studies described later found that 5/7 wells showed multiple mutations (Fig. 4A).

The reviewer is correct about the limitation of the PCR validation screen. A sentence clarifying this point has been added (lines 140-141), and the additional undetected mutations are now called out in the discussion (lines 253-254). Importantly, it was the 22/25 validation rate, which was lower than we had observed with other libraries constructed by our group, that caused us to do the deeper dive that allowed us to see the excess undetected mutations.

7. It's disturbing that of the three wells not containing predicted mutants (point 6.), only one could be further analyzed, but it was found to have two different unpredicted mutations. This indicates that many of the wells thought to contain single mutations likely carry undetected mutations.

Indeed, we were also concerned by this outcome, which is why we proceeded to investigate potential problems in the library. We had some early indications that the transposon was hopping, but follow-up experiments indicated the more likely explanation was mixed clones and undetected mutations. In response to this comment and comments from Reviewer 2, we have restructured the results to place the mixed clone data more prominently.

8. Line 77-80. I don't understand how the original *flaA* mutant was motile. Contamination with a second motile strain?

(N.B.: this was lines 177-80 in the original.) This section and accompanying Fig 3 were what led to our realization that the library contained mixed clones. Please note lines 169-170: "the *flaA* mutant was selected from a well of the library that was predicted to contain two additional transposons." This sentence is now edited for clarity. In Fig 3B, we show that these three transposons could be separated

into two isolates: one nonmotile isolate with an insertion in *flaA* and one motile isolate with two transposon insertions. A transition sentence has been added to clarify the investigation of mixed clones (lines 177-179).

9. line 205- given the extensive cross-contamination of wells, it's too bad the mutants in the condensed library weren't single colony purified. Purification and re-sequencing would have made this a much more useful resource.

Unfortunately, the scope of the mixed clone problem was not identified until after the condensed ordered library had been constructed. We don't believe that single-colony purification, sequencing, and re-arraying of 1728 mutants is a constructive use of our resources. In addition, as we note on lines 79-81, the condensed library is only one of many possibly useful subsets of the full library. We anticipate that effective screening of the library for phenotypes can be conducted with proper vetting of the positive hits (lines 349-368).

10. Will strains we distributed to the research community? If so, instructions for obtaining them should be included.

We will of course distribute strains to the research community upon request by contacting the corresponding author, per standard policy for scientific publication, and as we have done for our group's published *E. coli* and *K. pneumoniae* libraries (PMID 32358013 and 33720976, respectively). We would be delighted to deposit the full library in a repository; unfortunately, we have not identified any repositories that accept ordered libraries. Indeed, we would appreciate suggestions.

Reviewer #2 (Comments for the Author):

Review of Pearson et al., "Construction of an ordered transposon library for uropathogenic *Proteus mirabilis* HI4320"

Overview:

The authors use a new methodology to produce a structured transposon library in a model organism for urinary tract infections (a first example in *Proteus*?). This manuscript provides data supporting the library and discusses the gains/pitfalls of the library construction. The library could help initial mutagenesis screens for field-specific researchers. For the reader, most informative are the challenges and suggestions for future library constructions using these methods. However, it was difficult to parse this manuscript, and the most helpful information is missing emphasis and details. I've separated comments into major and minor below.

Although transposon mutagenesis has been used to study *P. mirabilis* for about 30 years, as cited in one of the comments below, to our knowledge, this study describes the first ordered transposon library for *P. mirabilis*. The methodology we used to generate the library was developed by others (references 11, 12, 17, and 37). *Below we answer specific critiques.*

Major:

I strongly encourage the authors to state what they think is the cause of the library problems more

explicitly. Start with the condensed library as a solution and then explain their rationale using the challenges as examples. The current manuscript ends with a final product (the condensed, structured transposon library), but most of the data speak to challenges in the initial version. Early statements in the text that seems like facts (e.g., lines 88 - 91) reflect the "draft" library. The reader can come away feeling that the statistics for "draft" (or initial) library (with errors) are the final ones. Then, the last portion of the manuscript reveals the final (usable to others) product and that the draft library is not usable. As stated above, I recommend that the authors focus first on the final library and its statistics and then do a retrospective framing of the challenges and caveats of this process. Here are some detailed suggestions:

The full ordered library consisting of 18k+ mutants, is neither a "draft" nor unusable. It is an ordered transposon library that can be divided into sub-libraries as convenient. For example, we have generated a condensed library that contains a single predicted transposon insertion per gene coding sequence. Issues we identified in the full library, such as uncalled mixed clones, will still be present in the condensed library (e.g., Fig. 4). We consider the full library to be a final product. So is the condensed library, which is one of many potentially useful subsets of the full library. Alternative condensed libraries could include, for instance, insertions in sRNAs, specific domains of proteins, or promoter elements; all of these would be similarly re-arrayed from the full ordered library. We have added some text clarifying the context of the condensed library (lines 79-81, 221-225). A new subheading has been added to the Discussion clearly stating that we think the major problems with the library stemmed from the mating and plating protocols (line 304).

- From this reader's perspective, the heart of the issue with the draft library seems to be how the initial colonies were chosen. The presented data points to a mistake in the initial arraying of the library, leading to some wells having mixed clones with varying fitness. Potential causes include the following: Yes, after considerable troubleshooting, we concluded that initial transposon mating conditions and colony picking were major sources of error in library construction. This is now further highlighted with a new subheading in the discussion (line 304).

- o The colonies were chosen with a colony picker rather than by hand. See lines 230 - 242: The study by Vandewalle et. el., 2015, used hand-picked colonies and showed no signs of mixed clones in their validation process. The follow-up study referred to a new library they did. They mentioned that picking five to ten clones from the glycerol stocks of the library was sufficient to recover the mutant in a clonally pure form. This step is a common good microbiology practice as some frequency of picking multiple clones to the same well is unavoidable during ordered library preparation (due to strong clumping behavior). For lines 240 - 242, the comparison should reflect manual or robot picking. Discuss the technical methods earlier in the manuscript.

We consulted with Kristof Vandewalle when figuring out how our *Proteus* library construction had gone wrong (noted in the Acknowledgements section). Vandewalle *et al* did indeed struggle with mixed clones in the original citation mentioned above. The direct quote provided states they needed to screen 5-10 colonies from their glycerol stocks to obtain the expected mutants. This is because the glycerol stocks contained mixed clones. Importantly, this issue was apparently less extensive in our previously published *E. coli* and *K. pneumoniae* libraries because we did not have to do these extra 5-10 colony picking and screening steps to identify mutants.

The same group cited problems with robot vs hand-picking in the second library (Borgers 2020) where mixed clones were even more prevalent (11 or 14% of wells contained mixed clones in each library). Unfortunately, this 2020 report about the method of colony picking was published after we had constructed our library.

o Perhaps there was some swarming on the initial plates. The authors used 1.5% LB agar, which seemed to allow for partial swarming in their hands (Figure S3B) and was reported to support swarming most recently in [Little K, Austerman J, Zheng J, Gibbs KA. Cell Shape and Population Migration Are Distinct Steps of *Proteus mirabilis* Swarming That Are Decoupled on High-Percentage Agar. *J Bacteriol.* 2019 May 8;201(11):e00726-18. doi: 10.1128/JB.00726-18. PMID: 30858303; PMCID: PMC6509654.]

For the strain in our manuscript, HI4320, 1.5% agar combined with 0.5 g/L NaCl is sufficient to prevent swarming. We did not observe swarming on the initial transposon mating plates. In contrast, the “partial” swarming mentioned in Fig. S3B was induced deliberately by spread-planting of diluted cultures to obtain single colonies on swarm agar (10 g/L NaCl), as stated in the figure legend. A new label has been directly applied to the figure specifying that swarm agar was used for this experiment.

We have personally encountered isolates that are either considerably more or less aggressive in swarming compared with HI4320, and the parameters to inhibit swarming by the strain in Little *et al* (that is, BB2000 and mutant derivatives) may well be different than they are for HI4320. Modulation of agar concentration to control these strain-to-strain differences in swarming motility has been widely reported, as reviewed in Pearson 2019 PMID 8932309 and going back to at least 1943 in Hayward and Miles, *Lancet*, where as much as 6-8% agar was required to prevent swarming by some strains.

• Regarding construction of the ordered library:

o Unclear is the basis for the model for determining the percent coverage of the transposon mutagenesis methodology. Please add clarifying comments to lines 76 - 85 or the methods. Vandewalle 2015 modeled expected transposon saturation of their transposon library in *M. bovis*, and we adapted the algorithm for use in *P. mirabilis*. The model involves a simulation of non-essential genomic regions (roughly, “genes”) that contain the average number of TA sites per gene (for *P. mirabilis* HI4320, we calculated an average of 76 TA sites per gene). The parameters for the coupon collector’s model and specific numbers we calculated for *P. mirabilis* HI4320 are stated in the methods, in a section now renamed “calculation of predicted genome saturation.” Additional details have been added there (lines 451-462), and the model has been set up more clearly in the results (lines 96-103).

♣ What were the parameters for the 'coupon collector' model? Will there be differences between libraries? Is it based on TA sites in the genome?

Please see previous comment. All libraries constructed with the same bacterial strain and a TA-inserting transposon will have the same prediction for the number of mutants required for saturation.

♣ The model predicts that with 18,432 mutants, 99.1% of the non-essential genes should be hit, which is above what the data set shows. Can you address this issue in the text? You assume that doubling the number of mutants will allow disrupting almost all targetable genes in this genome, yet this is not the case. Please discuss.

Lines 146-148 (emphasis added): “To address the lower-than-expected genome coverage and the many wells predicted to contain multiple transposon insertion events, we designed a series of experiments to explore potential complications with the *P. mirabilis* ordered library.” Indeed, much of the manuscript discusses this issue in some detail. In particular, please see lines 304-322.

♣ The TA sites do not seem to be a likely cause of the draft library problems. See the above discussion of the arraying.

We agree, which is why we state this was not likely to be a problem for *P. mirabilis* (lines 300-301), in contrast to prior libraries in other species. Because TA availability had been cited as a previous contributor to genome coverage, we mentioned the possibility before dismissing it. This section has now been shortened.

• Figure 1 is for the draft library, but the reader subsequently learns that a portion of the wells is incorrect. The authors do not provide an equivalent image for the final (“condensed”) library; table S3 is insufficient. Instead, consider moving Figure 1 to the supplemental figures.

The full library is not a “draft,” as noted above. Fig. 1 is distribution of transposons across the entire genome and is an accurate picture of the distribution of bioinformatically predicted transposons present in the full library. The condensed library, in contrast, consists of a single insertion per gene; a figure showing these single selected insertions would not be an accurate representation of the random distribution of transposons. The features of the condensed library are in Table 2 (previously Table 3) where we state that 1728 (45%) of ORFs were disrupted with single insertions identified. Indeed, Table S5 (previously Table S3) conveys the specific genes and precisely located transposons in the condensed library but was not meant to serve as an overview.

• Consider lines 132 - 142; this passage undercuts the entire library, raising questions as to why the draft library was discussed up until now.

The full (not “draft”) library’s details are essential to assessing sequencing metrics and transposon distribution. It contains transposons in sites that might not be represented in the condensed library, as discussed on lines 364-368, and thus is useful in instances where the condensed library is not. Importantly, even though every well in the condensed library should have a uniquely mapped *Proteus* mutant in it, we would expect a similar proportion of wells to have uncalled mixed clones or mutants that don’t correspond to the prediction.

o Also, the other two “new” mutations are in PMI_RS18625 (DNA topoisomerase) and PMI_RS00680 (RecR). The authors should include this information instead of requiring readers to find the information. Do the authors suspect certain gene regions are susceptible to incorrect insertions? Is there unexpected homology?

We regret the reviewer has mis-identified these insertions, which are not in genes encoding DNA topoisomerase or RecR. We have now added the predicted gene product to the text for one of these insertions (lines 152-154, *gshA*). The second mutation was not in a coding sequence (line 154, where it was specified as an intergenic region); we suspect it outgrew other mixed clones because it would not be predicted to have a fitness defect.

A major reason we endeavored to identify the transposon locations in the non-validating wells was because we were concerned about hotspots or transposon instability. However, we identified no such patterns and have no reason to suspect that certain gene regions are susceptible to incorrect insertions.

- Much of the text in the discussion considers the draft library; however, that tool is not usable to the research community. Please consider re-writing the discussion to mainly focus on the condensed library, as that is the product supporting this manuscript's publication.

Please see above; we disagree that the full library is not usable and we believe that it is essential to the manuscript. The full library contains insertions in regions that might be important even if they are not in coding regions; also, wells with multiple mapped insertions may be purified to extend genome coverage, as we showed for *flaA* in Fig. 3B. However, it is essential to understand the condensed library also contains unknown mixed clones, and the caveats we discuss apply to both the condensed and full libraries. To this end, we included sections in the discussion with our recommended best practices for using these libraries, whether using the full or condensed versions (lines 252-265 and 349-368). The condensed library's main advantage is that every well was bioinformatically predicted to have a single transposon insertion mapped to a coding sequence in the *P. mirabilis* genome. Thus, we believe it important to retain this pertinent text.

Some relevant references are missing:

(a) The authors cite transposon libraries in *P. mirabilis* strain HI4320. Yet, several groups published mutagenesis libraries in other strains (see below). A robust consideration of the challenges/constraints in those libraries is important as a justification in the introduction. Potential references are:

- Belas R, Erskine D, Flaherty D. Transposon mutagenesis in *Proteus mirabilis*. *J Bacteriol.* 1991 Oct;173(19):6289-93. doi: 10.1128/jb.173.19.6289-6293.1991. PMID: 1655704; PMCID: PMC208382.
- Stevenson LG, Rather PN. A novel gene involved in regulating the flagellar gene cascade in *Proteus mirabilis*. *J Bacteriol.* 2006 Nov;188(22):7830-9. doi: 10.1128/JB.00979-06. Epub 2006 Sep 15. PMID: 16980463; PMCID: PMC1636314.
- Gibbs KA, Urbanowski ML, Greenberg EP. Genetic determinants of self identity and social recognition in bacteria. *Science.* 2008 Jul 11;321(5886):256-9. doi: 10.1126/science.1160033. PMID: 18621670; PMCID: PMC2567286.
- Peters JM, Koo BM, Patino R, Heussler GE, Hearne CC, Qu J, Inclan YF, Hawkins JS, Lu CHS, Silvis MR, Harden MM, Osadnik H, Peters JE, Engel JN, Dutton RJ, Grossman AD, Gross CA, Rosenberg OS. Enabling genetic analysis of diverse bacteria with Mobile-CRISPRi. *Nat Microbiol.* 2019 Feb;4(2):244-250. doi: 10.1038/s41564-018-0327-z. Epub 2019 Jan 7. PMID: 30617347; PMCID: PMC6424567.

The manuscript is an announcement of an ordered library in *P. mirabilis* using a previously published method for identifying transposon insertions when mutants are arrayed in 96-well plates (Vandewalle 2015). Although the works cited above (and additional transposon mutagenesis studies not listed) are valuable to the *P. mirabilis* field, the current manuscript is not meant to provide a comprehensive review of transposon mutagenesis in *P. mirabilis*. We believe that adding this extra introductory material is not necessary to aid the reader's understanding of the manuscript.

We referenced two published Tn-seq libraries in *P. mirabilis* because they were directly relevant to analysis of our results. These studies (PMID 28614382 and 31009518, cited as references 29 and 30) used the same *mariner* transposon in the same strain of *P. mirabilis*, similarly relied on high throughput sequencing, and contained a saturating input number of transposons (50k). The mating protocol we

used came directly from these studies. Therefore, a direct comparison of our sequencing metrics with these particular published Tn-seq libraries was warranted in the results.

Importantly, the last reference in this list (Peters *et al*) showed the potential for knockdown in a variety of bacterial species, but the process was very low efficiency for *P. mirabilis*. The authors showed proof of concept for one gene in *P. mirabilis*, and stated that efficiencies in the range they observed for *P. mirabilis* were “more suited for single-gene knockdown approaches.” We have added a mention of this technology as a possible future direction in the discussion (lines 346-348).

(b) References and consideration of equivalent, genome-scale deletion libraries are missing (lines 39 - 41). Discussing these references would help this paragraph. Missing is the broad scope and something about the limitations or other disadvantages this study tries to overcome (cost, time-consuming?). Potential references are:

- Koo, B.-M. et al. Construction and analysis of two genome-scale deletion libraries for *Bacillus subtilis*. *Cell Syst.* 4, 291-305.e297 (2017).
- Baym, M., Shaket, L., Anzai, I. A., Adesina, O. & Barstow, B. Rapid construction of a whole-genome transposon insertion collection for *Shewanella oneidensis* by Knockout Sudoku. *Nat. Commun.* 7, 13270 (2016).
- Porwollik, S. et al. Defined single-gene and multi-gene deletion mutant collections in *Salmonella enterica* sv Typhimurium. *PLoS ONE* 9, e99820 (2014).

The Koo and Porwollik references, and additional notes about the feasibility of such feats, have been added to the introduction (lines 41-44). Importantly, the already-cited Keio collection is also a genome-scale deletion library.

The Baym reference is not a genome-scale deletion library, but instead is a different method of identifying locations of transposon mutants in an ordered library. This reference is now cited along with CP-CSeq as a useful algorithm for decreasing sequencing costs associated with identifying transposon insertions in arrayed libraries (lines 59-61).

The consideration of the bottleneck effect is confusing as written. This limitation appears due to all transposon libraries, not just those constructed with CP-Cseq. If retaining, consider expanding on the term "bottleneck issue" and put in the context of the larger field. How did your study address this limitation? How is it different (better?) than other approaches (consider the above references for additional approaches)? Please directly address these concerns or reframe the "bottleneck effect" limitations in the revised text, both in the introduction and discussion. Lines 241 and 242 should be edited to clarify the authors' points.

Bottleneck effects are a problem due to constraints specific to a given screening method, not the library itself. Smart library construction can reduce the number of mutants needed in a screen while maintaining good coverage. Yes, mutating every single gene one at a time would be appealing if feasible (Introduction, lines 41-42). For most labs, this is not. For *Proteus*, this would be incredibly difficult with currently developed protocols (*e.g.*, targetron or allelic exchange mutagenesis). Bottlenecks are clarified and discussed on lines 22-24, 53-56, and 263-265.

The paragraph starting at line 122 would benefit from more clarity. For example:

- Line 122: "Anchored PCR"? Please clarify the terminology, as this technique is standard.

This validation PCR step was described and cited in the methods, where we provided the name of the transposon-specific primer in the text (CP7) and showed the sequences in supplemental table S4. We have removed the term “anchored” and now placed this sentence in the methods more prominently under its own new heading (lines 463-474).

- Line 125: explicitly state that the insertion was confirmed.

The original text explicitly stated “PCR, using one gene-specific primer and one primer that reads outbound from both ends of the transposon, confirmed the transposon insertion in 22/25 of these mutants” (lines 137-139).

- Lines 129 - 131 reference "unexpected" results from libraries, but the "untraceable reads" and "many wells predicted to contain multiple transposon events." Yet, the text and the numbers presented do not match the urgency of this language or reveal the assays. Just earlier in the paragraph, the authors state that "the vast majority of the mutants [they] tested were validated" (emphasis is mine) but that the 3/25 was enough to trigger a broad introspection. These statements (and the distinction in urgency and concern) appear contradictory.

For wells *where mutants were mapped*, most of them confirmed (22/25), although not as many as in our previous two libraries. This means that most mapped mutants can be pulled from the library. The combination of observations, described earlier and brought together in this paragraph, prompted us to investigate the library more closely. This section has been edited for clarity. New text clarifying that validation of insertions does not exclude additional, uncalled insertions, has been added (lines 140-141).

- Separately, the condensed library is a fraction of the draft library. The authors end with a cautionary note about using the condensed library as well. So, is it really the "vast majority" of mutant strains that are valid to use?

Yes, the vast majority of the mutants are usable if one takes the precautions that we lay out in the discussion, which includes isolation and verification of a single colony-passaged mutant (lines 252-265). This could be done beforehand for a small number of specific mutants, or after a large primary screen, depending on experimental need (lines 350-369). Ultimately, as with any transposon screen, we would recommend follow-up work to validate the results, including genetic complementation and reconstruction of pertinent mutants using a directed mutagenesis method.

Lines 143 - 163 regarding native transposases and transposon stability:

- The presented data supports mixed clones (particularly given the colonies in Figure 3). It is unclear why the first discussed approach is to look at potential transposon hopping. The authors should consider moving the discussion of native transposases and transposon stability later or relegating it to the supplemental information.

Yes, we agree mixed clones are the more prominent issue (lines 162-204). We have restructured the Results to downplay instability by shortening and moving the instability section toward the end (lines 205-220) and adding text to emphasize mixed colonies (lines 140-141, 177-179).

- It seems that if native transposases are the cause, this would occur after library construction. Did the authors look at the stability of the condensed library strains, such as those in Figure 4? When the

colonies were repeatedly *freshly* taken from the frozen library, did strains from the condensed library retain the mapped insertion site? If so, please report this data.

Ten passages of *flhD* and *flaA* mutants did not result in transposon hopping from either locus (lines 170-175). Combined with the transposase deletion experiment, the results showed that transposon instability is much less likely to be the source of the library issues compared with uncalled mixed clones.

- Lines 148 - 150 argue for native transposases acting on the mariner transposon, which would be quite different from other bacteria for which mariner is used. Please include an alignment of 45 transposases with mariner transposases (either sequence or predicted structures) to support this assertion if retained.

As noted above, we have further downplayed instability by rephrasing these experiments and moving them later. The mention of 45 native transposases has been removed. There is no alignment of transposases to show because there were no hits, as we stated in the text (emphasis added): “BLAST of either the protein or nucleotide sequences of the transposase encoded by pSAM_AraC against the HI4320 genome **did not yield any hits**, so we predict the native transposases act on different lineages of mobile genetic elements” (lines 217-220).

- In addition, a scar is not guaranteed (as acknowledged in line 149 by the authors). [Note: clarify that "scar" refers to nucleotide traces of the insertion element.] As such, a single site is not sufficient to prove or disprove the hypothesis. If the authors choose to keep this section in a revised manuscript, additional mutant strains (that have new insertion sites) should also be sequenced.

The sentences about scars have been removed.

- Please clarify (regarding line 145) how many transposases are in the genome of the other two referenced libraries.

The other two ordered libraries published by our group (*E. coli* and *K. pneumoniae*) did not have the validation problems we encountered with the *P. mirabilis* library. Because these are entirely different species with their own lineages of transposable elements and transposases, we believe this comparison is not beneficial for the manuscript.

- Please state the number of colonies (line 156).

The number was approximate because there were too many colonies to count on the positive control plate (that is, the plate containing a mating experiment using the normal transposase induction protocol). Here are photos of the relevant plates from one of the mating experiments:

Some statements are too broad, given the presented data. For example (not comprehensive):

- There were saturation differences between the chromosome and plasmid; however, the text stated that coverage was similar. Please edit lines 88 - 91 to reflect.

Saturation was comparable between the chromosome and plasmid. The plasmid in Fig. 1 was not drawn to scale, which could have given the impression of reduced saturation. We have now redrawn the plasmid to more accurately reflect transposon insertion spacing, although it is still not to scale.

- Lines 120 - 121: "This method was effective at ensuring diverse genomic and library coverage." Yet, the following several sections speak to the many inaccuracies in the library, causing the formation of a "condensed" library. The condensed library has significantly less coverage in the plasmid and is not "diverse" in coverage (see Table S3).

The library issues did not cause the formation of a condensed library. The condensed library was a tool we planned to build from the outset to facilitate studies of single gene mutations (now clearly stated on lines 79-81 and 221-225). Fig. 1 shows broad genomic coverage of transposon insertions. Instead, the problems we encountered were in mapping transposon insertions to wells, insertion of multiple transposons in a single clone, and in the purity of clones in each well. Regarding plasmid coverage in the condensed library, 25/47 ORFs (53.2%) are represented, which is in fact slightly better coverage per ORF than the chromosome.

- The statement in line 162 ("colony contamination is very easy to obtain") is overly broad and applicable to every bacterium.

Based on our experience and other published ordered libraries, there is a persistent issue with picking colonies for ordered libraries. We are hard pressed to think of more different bacteria than *Proteus* and *Mycobacteria*, yet we had similar problems as Vandewalle/Borgers with colony picking. The prevalence of mixed clones in wells that were only supposed to obtain single picked colonies was a major outcome for our study, and we believe it is important to call attention to this result (line 216).

Data availability:

- Please update Table S3 (condensed library) to better match equivalent studies, such as the detailed supplementary in [Vandewalle K, Festjens N, Plets E, Vuylsteke M, Saeys Y, Callewaert N.

Characterization of genome-wide ordered sequence-tagged *Mycobacterium* mutant libraries by Cartesian Pooling-Coordinate Sequencing. *Nat Commun.* 2015 May 11;6:7106. doi: 10.1038/ncomms8106. PMID: 25960123; PMCID: PMC4432585].

- Minimally, include all transposon insertion events in the condensed library.

This table (now Table S5) solely consists of all mapped transposons in the condensed library. Precise transposon insertion loci are listed in column C. We have now added gene names and annotations.

Minor

The manuscript frames comparisons to *K. pneumoniae* and *E. coli*, but the rationale for this process is unclear, especially since that data appears to be separately published.

A new mention that the *K. pneumoniae* and *E. coli* libraries are published has been added to lines 120-121. We cited these published studies both in text and tables because they show construction of ordered libraries by our group that used the same CP-CSeq methodology to map insertions to library wells. We used these studies as a comparison because we had very different results and technical difficulties for *P. mirabilis* than for these other libraries made by our group.

Consider adding header sentences.

New boldface headers (lines 105, 304, 425, 463) have been added to complement pre-existing header sentences throughout the results and discussion, e.g., lines 83, 132, 149, etc.

Present the data as percentages. For example, X% of the transposon insertion event happened within ORF (Figure 2C). How many genes remain unhit?

The original text stated: “our library contains disruptions in 1728 open reading frames, of ~3812 total between the chromosome and plasmid (45%) (Fig. 2C).” We have now moved the percentage immediately after the absolute number (line 126) and added percentages to additional lines (113-114, 126). Fig. 2 shows percentages and the total number (n) under each panel. Finer details on raw numbers are shown in Table S2 (previously Table 1).

Figure 3 and lines 182 - 184, what is the band in lane 7 that is absent from the other lanes?

The extra band could be dismissed as incomplete digestion of genomic DNA, a transposon hopping event, or a contaminant. Given the very slight band just above the slowest migrating band in this lane, we are inclined to think it is a digestion issue, although this was not apparent from the appearance of the digested DNA (below). Either way, it is clear that the three original transposon insertions separate into two lineages (lanes 3, 5, and 8 vs lanes 4, 6, and 7). A comment about this band is now on line 206.

November 1, 2022

Dr. Melanie M Pearson
University of Michigan Medical School
Microbiology and Immunology
5641 Medical Science Bldg II
1150 W Medical Center Dr
Ann Arbor, MI 48109-0620

Re: Spectrum03142-22R1 (Construction of an ordered transposon library for uropathogenic *Proteus mirabilis* HI4320)

Dear Melanie,

Your manuscript has been accepted, and I am forwarding it to the ASM Journals Department for publication. You will be notified when your proofs are ready to be viewed.

Sincerely,

Philip Rather
Editor, Microbiology Spectrum

Journals Department
Supplemental Tables S1-S7: Accept
Supplemental Material: Accept